# Increased localization of APP-C99 in mitochondria-associated ER membranes causes mitochondrial dysfunction in Alzheimer disease

Marta Pera[1,†], Delfina Larrea[1], Cristina Guardia-Laguarta[2], Jorge Montesinos[1], Kevin R Velasco[1], Rishi R Agrawal[3], Yimeng Xu[2], Robin B Chan[2], Gilbert Di Paolo[2,‡], Mark F Mehler[4], Geoffrey S Perumal[5], Frank P Macaluso[5], Zachary Z Freyberg[6], Rebeca Acin-Perez[7], Jose Antonio Enriquez[7], Eric A Schon[1,8] & Estela Area-Gomez[1,*] (ID)

## Abstract

In the amyloidogenic pathway associated with Alzheimer disease (AD), the amyloid precursor protein (APP) is cleaved by β-secretase to generate a 99-aa C-terminal fragment (C99) that is then cleaved by γ-secretase to generate the β-amyloid (Aβ) found in senile plaques. In previous reports, we and others have shown that γ-secretase activity is enriched in mitochondria-associated endoplasmic reticulum (ER) membranes (MAM) and that ER–mitochondrial connectivity and MAM function are upregulated in AD. We now show that C99, in addition to its localization in endosomes, can also be found in MAM, where it is normally processed rapidly by γ-secretase. In cell models of AD, however, the concentration of unprocessed C99 increases in MAM regions, resulting in elevated sphingolipid turnover and an altered lipid composition of both MAM and mitochondrial membranes. In turn, this change in mitochondrial membrane composition interferes with the proper assembly and activity of mitochondrial respiratory supercomplexes, thereby likely contributing to the bioenergetic defects characteristic of AD.

**Keywords** Alzheimer's disease; C99; MAM; mitochondria and sphingolipids
**Subject Categories** Membrane & Intracellular Transport; Metabolism; Neuroscience
**The EMBO Journal (2017) 36: 3356–3371**

## Introduction

Familial AD (FAD) is characterized by mutations in presenilin-1 (PS1), presenilin-2 (PS2), and amyloid precursor protein (APP). APP is first cleaved by either α-secretase or β-secretase (BACE1) to produce C-terminal fragments (CTFs) 83 aa (C83) or 99 aa (C99) long, respectively. PS1 and PS2 are the catalytic subunits of the γ-secretase complex that cleaves C83 and C99 to produce either p3 or β-amyloid (Aβ; ~40 aa), respectively, along with the APP intracellular domain (AICD). The accumulation of Aβ, and especially its longer forms (e.g., ~42 aa), within plaques, together with Tau tangles, are the neuropathological hallmarks of AD. The deleterious effects of Aβ deposition during the symptomatic stages of AD are undeniable (Hardy & Higgins, 1992), but the role of Aβ in earlier phases of the disease is still debated.

During these early stages, AD cells exhibit alterations in numerous metabolic processes (McBrayer & Nixon, 2013; Wang *et al*, 2014). Among these, perturbed mitochondrial function, including reduced respiratory chain activity and ATP production, and increased oxidative stress (Du *et al*, 2010), have been described extensively (Swerdlow *et al*, 2014), occurring before the appearance of plaques (Yao *et al*, 2009; Wang *et al*, 2014). Nevertheless, the cause of the mitochondrial deficits in AD is still unknown.

In addition to mitochondrial dysfunction, alterations in lipid metabolism are another feature of AD (Mapstone *et al*, 2014), but their origin and relationship to APP metabolism are unclear. Among these alterations, abnormal sphingolipid metabolism has been reported in AD tissues (van Echten-Deckert & Walter, 2012). Specifically, there is an upregulation of *de novo* ceramide synthesis

---

1  Department of Neurology, Columbia University Medical Center, New York, NY, USA
2  Department of Pathology and Cell Biology, Columbia University Medical Center, New York, NY, USA
3  Institute of Human Nutrition, Columbia University Medical Campus, New York, NY, USA
4  Departments of Neurology, Neuroscience, and Psychiatry and Behavioral Sciences, Albert Einstein College of Medicine, Bronx, NY, USA
5  Analytical Imaging Facility, Albert Einstein College of Medicine, Bronx, NY, USA
6  Departments of Psychiatry and Cell Biology, University of Pittsburgh, Pittsburgh, PA, USA
7  Cardiovascular Metabolism Program, Centro Nacional de Investigaciones Cardiovasculares Carlos III (CNIC), Madrid, Spain
8  Department of Genetics and Development, Columbia University Medical Center, New York, NY, USA
   *Corresponding author. Tel: +1 212 305 3836; Fax: +1 212 305 3986; E-mail: eag2118@cumc.columbia.edu
   †Present address: Cell Factory, Unit of Cell therapy and Cryobiology, Fondazione IRCC Ca' Granda Ospedale Maggiore Policlinico, Milan, Italy
   ‡Present address: Denali Therapeutics, South San Francisco, CA, USA

(Grimm *et al*, 2011) and an increase in the activity of sphingomyelinase (SMase), which catabolizes sphingomyelin (SM) into ceramide (Filippov *et al*, 2012). These alterations act synergistically to increase ceramide content in AD brains (Filippov *et al*, 2012).

As these metabolic alterations occur early in AD, they cannot be explained by the accumulation of plaques or tangles. Moreover, unsuccessful efforts directed toward modifying Aβ production as a treatment for AD (Castello *et al*, 2014) have raised the possibility that other aspects of APP cleavage may be contributing to these metabolic changes. In this regard, increased levels of the C99 fragment have also been shown to contribute to AD pathogenesis (Lee *et al*, 2006; Lauritzen *et al*, 2012), suggesting a role for C99 in the early stages of pathogenesis.

The processing of APP occurs in lipid raft domains (Cordy *et al*, 2006), which are membrane regions enriched in cholesterol and sphingolipids (Pike, 2009). While most of these domains are found in the plasma membrane, intracellular lipid rafts have also been described (Browman *et al*, 2006). One of these intracellular lipid rafts is called mitochondria-associated ER membranes (MAM), a functional subdomain of the ER located in close apposition to mitochondria that regulates key cellular metabolic functions (Vance, 2014).

We and others have shown that presenilins and γ-secretase activity localize to MAM (Area-Gomez *et al*, 2009; Newman *et al*, 2014; Schreiner *et al*, 2015). Moreover, MAM functionality (Area-Gomez *et al*, 2012) and ER–mitochondrial apposition (Area-Gomez *et al*, 2012; Hedskog *et al*, 2013) are increased in AD.

We now report that the concentration of unprocessed C99 at the MAM is increased in cell and animal models of AD and in cells from AD patients. This increase in MAM-localized C99 is associated with the activation of sphingolipid synthesis and hydrolysis, and with a subsequent increase in ceramide levels [notably, a feature observed in AD (Cutler *et al*, 2004; He *et al*, 2010), particularly in mitochondrial membranes (Kennedy *et al*, 2016)]. Finally, we show that these higher levels of ceramide on mitochondria cause reduced respiratory chain activity. Given these results, we propose that a critical component of AD pathogenesis is mediated by C99 toxicity through its effects on MAM and mitochondria.

# Results

## C99 inhibits mitochondrial respiration in presenilin-mutant cells

Current hypotheses regarding mitochondrial dysfunction in AD propose that this defect is the consequence of the accumulation of Aβ in mitochondria (Manczak *et al*, 2006), but the mechanism is unclear. To address this, we measured mitochondrial respiration in fibroblasts from FAD patients with pathogenic mutations in PS1 (M146L and A246E) and in age-matched controls, as well as in mitochondria from the brain of a knock-in (KI) mouse model expressing the M146V mutation in PS1 (PS-KI$^{M146V}$) (Guo *et al*, 1999). We observed reduced respiration in FAD patient cells (Fig 1A, and Appendix Fig S1A and P) and in mitochondria isolated from PS-KI$^{M146V}$ mouse brain (Appendix Fig S1B). To understand the consequences of presenilin mutations and the effect of amyloid on mitochondrial function in AD, we measured respiration in mouse embryonic fibroblasts (MEFs) ablated for both *Psen1* and *Psen2*

(PS-DKO) (Herreman *et al*, 2000). As above, we found decreased respiration in the PS-DKO cells compared to controls (Fig 1B and Appendix Fig S1Q). Additionally, measurements of oxygen consumption rate (OCR) in permeabilized cultures of PS-DKO cells showed clear defects in respiration (Appendix Fig S1D). Importantly, the decrease in respiration was not due to reductions in mitochondrial content or biogenesis (Appendix Fig S1C, F, G, and I). Taken together, these results suggest that, from the mitochondrial perspective, cells with pathological mutations in, and ablation of, presenilins behave similarly, resulting in loss of mitochondrial respiration. Given that PS-DKO MEFs lack γ-secretase, these results suggest that mitochondrial dysfunction in these mutant cells does not depend on Aβ production.

To determine whether mutations in presenilins affect mitochondria via its role as the catalytic core of γ-secretase, we measured mitochondrial respiration in human neuroblastoma SH-SY5Y cells treated with 10 μM of the γ-secretase inhibitor DAPT. This inhibition caused a significant reduction in respiration compared to that in untreated cells (Fig 1C and Appendix Fig S1R) without altering mitochondrial content or biogenesis (Appendix Fig S1F, I, and J). This result implies that the catalytic activity of presenilins is necessary to maintain respiratory function. In addition, given that neither PS-DKO nor DAPT-treated cells produce Aβ, our results raise the possibility that the mitochondrial deficits in AD are independent of Aβ production. However, it is equally possible that alterations in full-length APP (FL-APP) or in any of its cleavage products may play a role in regulating mitochondrial respiration. To test this, we measured oxygen consumption in MEFs in which *App* and its paralog *Aplp2* were knocked out (APP-DKO) (Zhang *et al*, 2013). Contrary to what we found in presenilin-mutant cells, elimination of APP and APLP2 had no detrimental effects on respiration (Fig 1D). In fact, the OCR in permeabilized APP-DKO cells was slightly but significantly increased compared to that in controls (Fig 1D and Appendix Fig S1E and T).

Considering that PS-DKO cells and APP-DKO cells both lack Aβ and AICD, our results suggest that the difference in mitochondrial function observed in these two cell models was due to the presence or absence of FL-APP or its cleavage products, C99 and C83. We therefore measured respiration in PS-DKO cells treated with BACE1 inhibitor IV (BI) (Fig 1E and Appendix Fig S1S) and with an α-secretase inhibitor (TAPI-1) (Appendix Fig S1H) to abrogate the production of C99 and/or C83, respectively. As controls, we added back physiological concentrations of Aβ and oligomers of Aβ$_{42}$ (Appendix Fig S1K–M). Remarkably, only the treatment with BI rescued the respiration defects, both in PS-DKO cells (Fig 1E and Appendix Fig S1S) and in FAD fibroblasts (Appendix Fig S1N), suggesting that increased levels of unprocessed C99, rather than the levels of Aβ, play a role in the mitochondrial dysfunction seen in AD. Supporting this, addition of Aβ oligomers to APP-DKO cells had little effect on respiration (Appendix Fig S1M), whereas APP-DKO cells expressing C99 suffered a significant decrease in respiration, which was accentuated by adding DAPT (Fig 1D and Appendix Fig S1T), without changes in the content of mitochondria (Appendix Fig S1O).

## C99 can be localized in MAM

APP and its cleavage products have been shown to colocalize with almost every membranous compartment in the cell, including

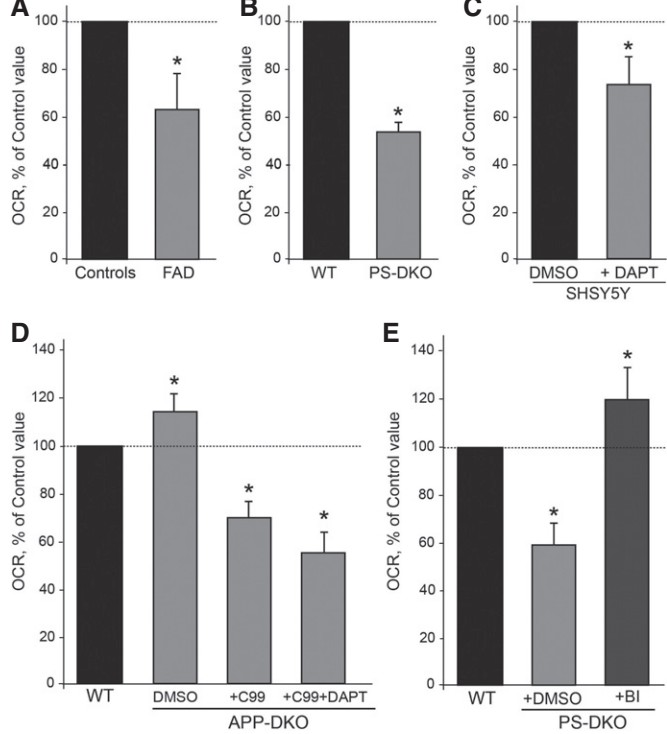

**Figure 1.  Mitochondrial respiration [oxygen consumption rate (OCR)] in γ-secretase-deficient cells.**

A   Oxygen consumption rate in AD fibroblasts (FAD).
B   Oxygen consumption rate in PS-DKO MEFs.
C   Oxygen consumption rate in SH-SY5Y cells treated with DAPT, an inhibitor of γ-secretase activity.
D   Oxygen consumption rate in APP-DKO MEFs before and after overexpression of C99.
E   PS-DKO MEFs treated with BACE inhibitor (BI).

Data information: Data represent averages of $n > 5$ independent experiments ± SD. *$P < 0.05$. Analysis by unpaired $t$-test.

mitochondria (Devi & Ohno, 2012). Thus, it is possible that in AD cells, C99, as previously suggested for Aβ (Casley et al, 2002), is retained on mitochondrial membranes, disrupting its regulation.

To explore the localization of C99 and C83, we isolated subcellular fractions from mouse brain (Area-Gomez, 2014) (Appendix Fig S2A) and analyzed them by Western blot [this was also validated in an identical fractionation of mouse liver (Appendix Fig S2B)], using specific markers for each compartment. Interestingly, we found that APP-CTF fragments, while present in all compartments, were enriched significantly in MAM regions of the ER (Fig 2A).

To discriminate between the localization of C83 and C99, we isolated subcellular fractions from SH-SY5Y cells treated with DAPT (to prevent cleavage of C99 and C83), or treated with α- and γ-secretase inhibitors (TAPI-1 and DAPT, to prevent the generation of C83 and the cleavage of C99, respectively, thereby revealing the presence only of C99). Notably, Western blot analysis of these fractions showed that while C83 was present in all samples (i.e., cells treated only with DAPT), C99 was located preferentially in MAM (i.e., fractions from cells treated with TAPI-1 and DAPT) (Fig 2B).

Many reports have shown that C99 is localized mainly in endosomes (Haass et al, 2012; Das et al, 2016). Therefore, it is possible that

the presence of C99 in MAM regions was the result of a cross-contamination of MAM samples with endosomes during the process of subcellular fractionation. To eliminate this possibility, we isolated cellular membranes from mouse brain and separated them through continuous density sucrose gradients (Appendix Fig S2C), to allow us to purify MAM away from endosomes and other subcellular fractions. After gradient centrifugation, we examined the distribution of C83 and C99 compared to markers for other compartments (Fig 2C). Consistent with the data of others (Das et al, 2016), FL-APP and BACE1 co-migrated partially with a marker for endosomes (Rab7), but not with lysosomal, ER-intermediate, or MAM markers (Fig 2C). Similarly, the APP-CTFs C83 and C99 co-migrated with endosomal and lysosomal markers (Rab5, Rab7, and LAMP-2) (Haass et al, 2012; Das et al, 2016), whereas PS1 co-migrated with MAM markers, such as FACL4 (Area-Gomez et al, 2009; Newman et al, 2014; Schreiner et al, 2015). We reasoned that the difficulty in seeing APP-CTFs and PS1 together was probably due to the rapid cleavage of the CTFs by γ-secretase once both are in the same compartment. Thus, to circumvent this rapid cleavage and determine C99 localization, we repeated the same analysis using PS-DKO cells (Herreman et al, 2000) in which APP-CTFs are not cleaved, due to the absence of presenilins (Appendix Fig S1G). Western blot analysis in this case showed that, in addition to its localization in endosomes, a significant fraction of unprocessed C99 co-migrated with MAM markers (Fig 2D).

To validate this result by imaging, we transfected wild-type (WT) MEFs and COS-7 cells with plasmids expressing fluorescently tagged C99 and mitochondrial and ER markers, and in the absence (Appendix Fig S2D) or presence (Fig 2E) of γ-secretase inhibitors. Confocal microscopy analysis revealed that C99 (in red) was present mainly in the cytosol and in ER membranes (in green), as shown previously by others (Das et al, 2016). In addition to those sites, C99 (in red) also colocalized with regions where both ER (in green) and mitochondria (in blue) were present (white arrows in Fig 2E and Appendix Fig S2D). This suggests that, like presenilins (Area-Gomez et al, 2009), C99 can be localized to areas of the ER apposed to mitochondria, that is, MAM, and is consistent with the fact that γ-secretase activity is present in this compartment (Area-Gomez et al, 2009; Schreiner et al, 2015).

To corroborate this result, we analyzed the localization of unprocessed C99 by immunogold electron microscopy (iEM) of PS-DKO cells, using antibodies against C-terminal regions of APP. In agreement with the confocal and Western blot analyses, iEM images indicated that, when uncleaved, C99 can be localized in MAM regions of the ER (Fig 2F and Appendix Fig S2E).

We next asked whether the increased localization of C99 in the MAM also occurred in the context of AD, as tissues from AD patients and animal models show increases in this fragment (Holsinger et al, 2002). We measured C99 levels by Western blot in homogenates from embryonic cortical neurons from WT and PS1-KI[M146V] mouse brain (Guo et al, 1999) (Appendix Fig S2F), as well as from cells from AD patients and controls (Appendix Fig S2G). Notably, there was more C99 in the homogenates of mutant neurons and of cells from AD patients, than in those from controls (Appendix Fig S2F and G), similar to previous findings in other AD patients and in FAD mice (McPhie et al, 1997; Yang et al, 2003; Rockenstein et al, 2005). In addition, Western blot analysis of subcellular fractions isolated from WT and PS-KI[M146V] mouse brain also showed an increase in the levels of C99 in the mutant samples,

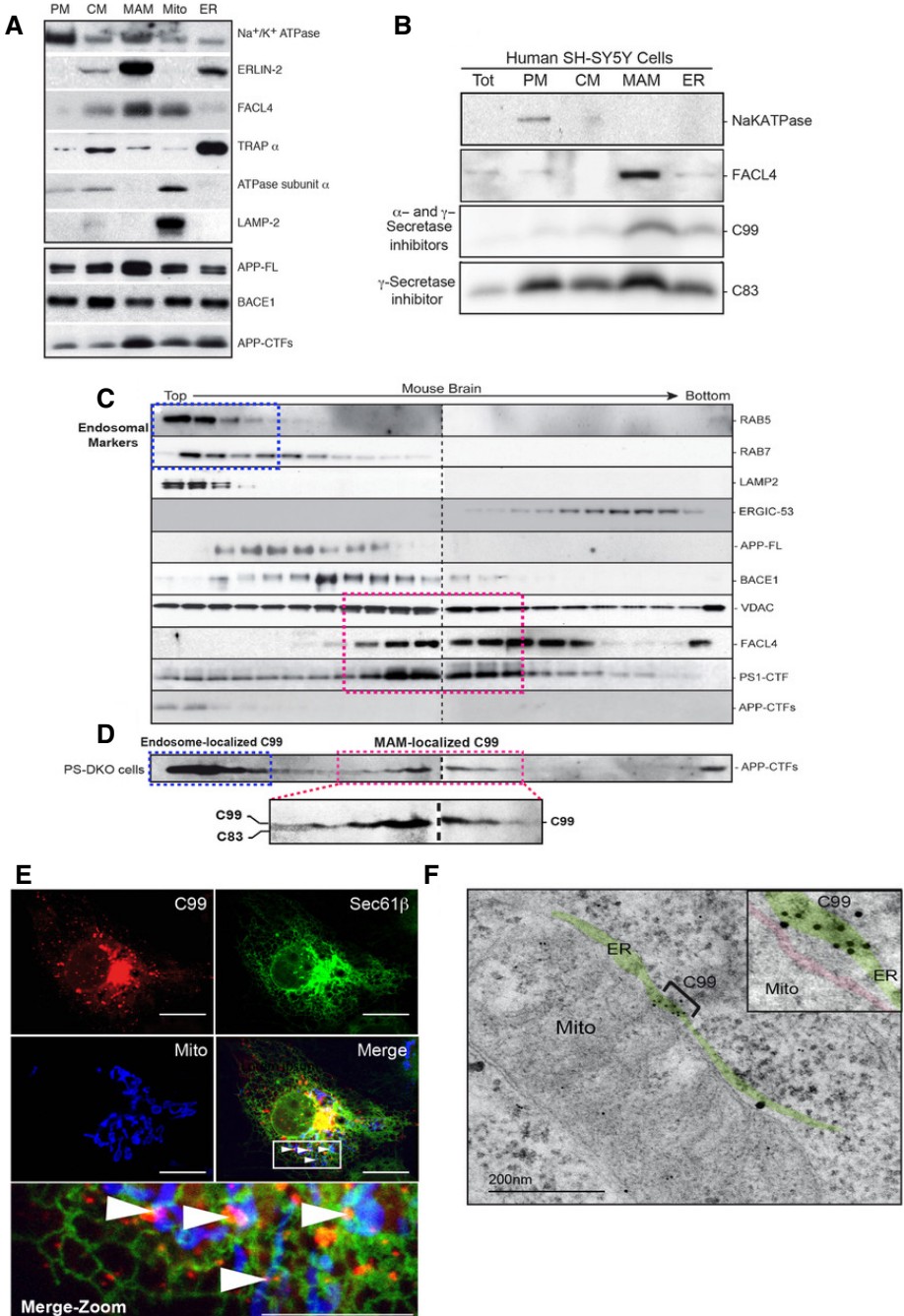

**Figure 2. Localization of C99 to MAM.**

A   Western blot of fractions from mouse brain probed with the indicated antibodies (30 μg of protein per lane). PM, plasma membrane; CM, crude membrane fractions.

B   Western blot of subcellular fractions from SH-SY5Y cells treated with α- and γ-secretase inhibitors to reveal the differential localization of APP-CTF fragments. Note that C99 is located predominantly in the MAM fraction (30 μg of protein per lane). Tot, Total homogenate.

C   The crude membrane fraction (CM) from mouse brain was treated with detergent, purified on a continuous sucrose gradient, and gradient fractions were analyzed by Western blot [two parallel gels (dotted line)], probing with antibodies to detect the indicated marker proteins. Blue dotted boxes indicate areas of the gradient enriched in endosomal and lysosomal markers. Pink dotted boxes indicate areas of the gradient enriched in MAM markers.

D   The same analysis using crude membrane fractions from PS-DKO MEFs to reveal the differential distribution of C83 vs. C99.

E   Representative confocal image of a DAPT-treated COS-7 cell shows that uncleaved C99 (red) and mitochondria (blue) can colocalize only in those areas where ER (green) is also present (i.e., MAM). Note how C99 colocalizes with areas where mitochondria and ER are apposed [white arrows in Merge (boxed), also shown in the expanded view (Merge-Zoom)]. Scale bars = 20 μm in top panels, 10 μm in bottom (Merge-Zoom) panel.

F   Representative immunoelectron microscopy image of PS-DKO cells incubated with antibodies against APP-CTF conjugated with immunogold particles, showing retention of C99 in MAM areas of the ER. The ER is colored in green, and the mitochondrial outer membrane in red. Note significant labeling in MAM regions (bracketed area, highlighted in inset).

and especially in the MAM fractions, compared to that in controls (Appendix Fig S2H), while the relative concentration of AICD was not changed significantly (Appendix Fig S2H).

Taken together, our results suggest that C99, after being produced in endocytic compartments (Das *et al*, 2016), is targeted to MAM, via an as-yet unknown mechanism, to be cleaved rapidly by γ-secretase. Moreover, both pathogenic mutations in PS1 and reductions in γ-secretase activity cause the retention of this fragment in this region of the ER that is in close apposition to mitochondria.

## Increased localization of C99 at MAM upregulates MAM functionality and ER–mitochondrial apposition

Given that reduced γ-secretase activity causes an accumulation of C99 at the MAM, we asked whether elevated C99 could be the cause of the increased ER–mitochondria apposition and MAM upregulation seen in AD (Area-Gomez *et al*, 2012). To assess apposition, we transfected control and PS-DKO cells with markers of ER and mitochondria, and measured their colocalization (de Brito & Scorrano, 2008; Area-Gomez *et al*, 2012) in the absence or presence of BI to prevent the generation of C99; remarkably, incubation with BI rescued the upregulation of ER–mitochondria apposition seen in mutant cells (Fig 3A and B).

To assess the effect of C99 on MAM functionality, we measured the conversion of cholesterol to cholesteryl esters by ACAT1, a MAM-resident enzyme (Area-Gomez *et al*, 2012), and monitored the accumulation of newly synthesized cholesteryl esters in lipid droplets (LDs) (Area-Gomez *et al*, 2012). Treatment with BI reduced the incorporation of cholesterol into cholesteryl esters (Fig 3C) and reduced the number of LDs in PS-DKO cells, in AD patient fibroblasts (Fig 3D), and in PS1-KI^M146V mouse astrocytes and cortical neurons (Appendix Fig S3A). Similarly, treatment of PS-DKO cells with Gleevec, an anticancer drug that has been recently shown to reduce APP cleavage by BACE (Netzer *et al*, 2017), resulted in a significant reduction of LDs in PS-DKO cells (Appendix Fig S3B). Lipid droplets also accumulated in SH-SY5Y and HeLa cells treated with DAPT alone (i.e., increasing C99), which was reversed in cells treated with DAPT+BI (i.e., preventing C99 formation) (Appendix Fig S3C). Supporting these data, and contrary to what we observed in γ-secretase-deficient cells but in agreement with what we saw in the DAPT+BI treated cells, MEFs in which *BACE1* had been knocked out (Luo *et al*, 2001) were essentially devoid of LDs in the cytosol (Appendix Fig S3D).

Taken together, these results show that the increase in, and retention of, uncleaved C99 in the MAM induces both a physical and functional enhancement of ER–mitochondria connections.

## Sphingolipid metabolism is perturbed in AD-mutant cells

Given that MAM is a lipid raft (Area-Gomez *et al*, 2012), we speculated that C99 could have a role in MAM activity and in ER–mitochondrial connectivity through changes in MAM lipid composition (Simons & Vaz, 2004). We therefore performed lipidomic analyses of total homogenates, mitochondrial fractions, and isolated MAM from PS-DKO MEFs and controls. We found a significant increase in ceramide (Fig 4A and Appendix Fig S4A, left panel) and a parallel decrease in sphingomyelin in mutant cells (Fig 4B and Appendix Fig S4B, right panel), which was more pronounced in the mitochondrial

(Fig 4A and B, and Appendix Fig S4B) and MAM (Fig 4C and Appendix Fig S4C) fractions than in total homogenates. To confirm the presenilin-dependent nature of these lipid alterations, we transfected the PS-DKO cells with plasmids expressing either PS1^WT or PS1^A246E (Appendix Fig S4D and E). Notably, expression of PS1^WT, but not PS1^A246E, was capable of partially rescuing the alteration in sphingolipid content in PS-DKO cells (Appendix Fig S4F).

Moreover, there was an inverse relationship between the amounts of individual sphingomyelin species present and those of the corresponding ceramide species (Appendix Fig S4G). The latter result suggested that there was an increase in the hydrolysis of sphingomyelin by sphingomyelinases (SMases) and subsequent upregulation of the *de novo* synthesis of SM to replace its loss (Fig 4D). In agreement with this idea, PS-DKO cells showed a significantly higher synthesis of both ceramide and sphingomyelin vs. WT (Fig 4D). In addition, acidic (aSMase) and neutral (nSMase) SMase activities were increased in the PS-DKO cells (Fig 4E), with a more dramatic upregulation of nSMase activity, correlating with increased expression of neutral sphingomyelinase 2 (nSMase2; gene *Smpd3*) (Appendix Fig S4H). We also observed increases in both acid and neutral SMase activities in PS1-KI^M146V mouse brain (Appendix Fig S4I). Likewise, we replicated the increase in SMase activity in SH-SY5Y cells by inhibiting γ-secretase activity (Appendix Fig S4J), suggesting that the effects of mutated presenilins on sphingolipid metabolism occur via their roles as proteases in γ-secretase. In agreement with this view and with our lipidomics results, expression of PS1^WT, but not PS1^A246E, significantly blunted the upregulation of nSMase activity in PS-DKO cells (Appendix Fig S4K).

To understand whether these effects were direct or were mediated by APP and/or its cleavage products, we measured SMase activity in APP-DKO cells. Contrary to what we found in PS-DKO and DAPT-treated cells, APP-DKO cells showed significant decreases in both sphingolipid synthesis (Appendix Fig S4L) and SMase activities (Appendix Fig S4M). As mentioned previously, both PS-DKO and APP-DKO cells lack Aβ and AICD. Therefore, any difference in sphingolipid regulation between the two cell types must be due to the presence or absence of full-length APP and/or C83 and C99. We therefore measured SMase activities in PS-DKO cells treated with α- and β-secretase inhibitors to test the effect of C83 and C99, respectively, as well as in PS-DKO cells in which Aβ and AICD were added back (Appendix Fig S4N). Interestingly, only the inhibition of C99 production (by BI) resulted in an attenuation of sphingolipid synthesis and hydrolysis by sphingomyelinases (Fig 4E). These results indicate that it is the increase in C99 that causes the upregulation of sphingolipid metabolism, resulting in the previously described elevations in ceramide in AD (Filippov *et al*, 2012). However, they did not clarify why ceramide is particularly elevated in MAM and mitochondrial membranes.

## MAM participates in the regulation of sphingolipid metabolism

Previous reports have suggested that MAM is involved in regulating sphingolipid metabolism, affecting mitochondrial activity (Ardail *et al*, 2003). In fact, mitochondria are reported to contain ceramide, probably generated at MAM (Kogot-Levin & Saada, 2014). Taking these and our data into account, we hypothesized that an increase in ceramide synthesis and in SMase activity at ER–mitochondria

**Figure 3.  ER–mitochondrial apposition is regulated by C99.**

A   Localization of ER (green) and mitochondria (red) in the indicated MEFs without and with BACE1 inhibitor (inhibiting C99 formation; see Western in panel B). Large boxes in the Merge are enlargements of the small boxes. Scale bars = 20 μm.

B   Quantitation by ImageJ analysis of the colocalization of ER and mitochondrial signals from experiments like the one shown in (A) (average of *n* = 4 independent experiments ± SD). *$P$ < 0.05. Analysis by unpaired *t*-test. The Western blot indicates the APP-CTF levels in the indicated cells (30 μg of protein per lane).

C   ACAT1 activity in WT and PS-DKO MEFs in the presence and absence of α-, β-, and/or γ-secretase inhibitors (average of *n* = 4 independent experiments ± SD). *$P$ < 0.05. Analysis by unpaired *t*-test. ACAT activity was normalized by controls (WT incubated with vehicle [DMSO]).

D   Staining of the indicated cells with LipidTox Green to detect lipid droplets. Scale bars = 20 μm.

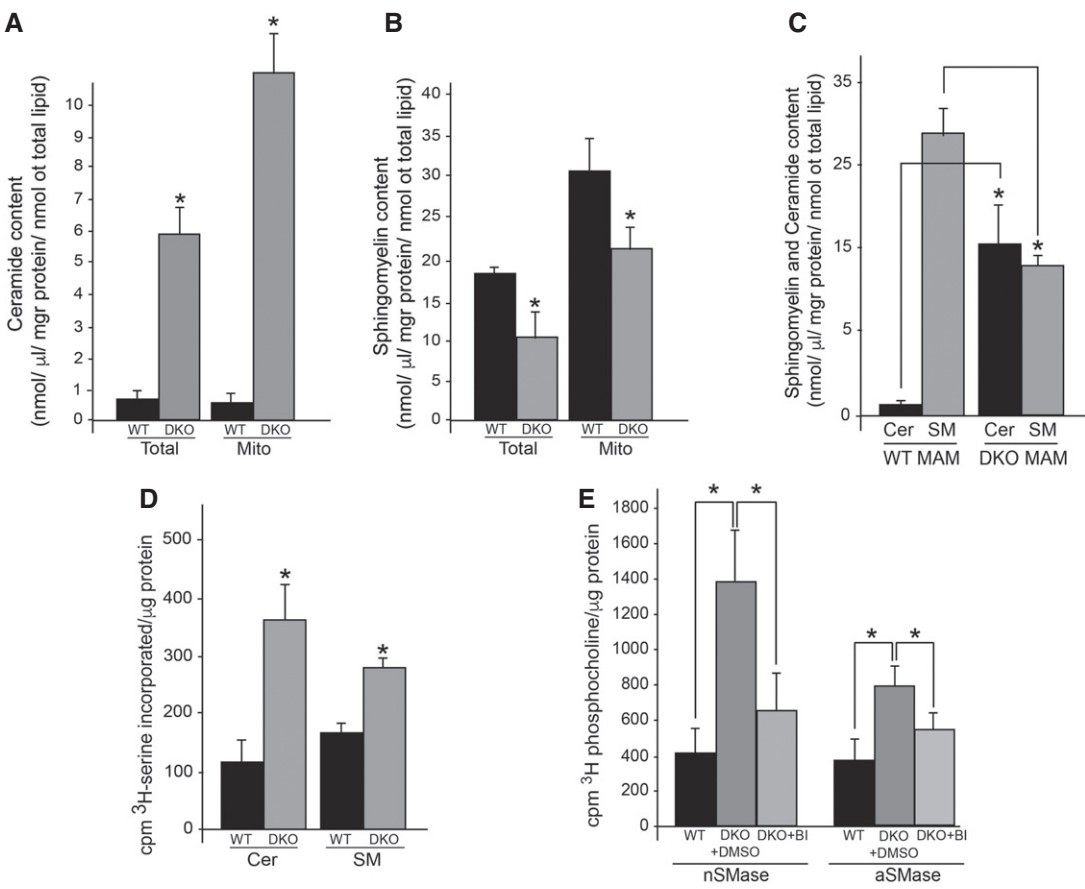

**Figure 4.  Sphingolipid metabolism in PS-DKO MEFs.**

A, B  Ceramide (A) and sphingomyelin (B) levels in total homogenate and in crude mitochondrial fractions in WT and PS-DKO MEFs. Lipid units are represented as molar mass over total moles of lipids analyzed (mol%) (average of *n* = 5 independent experiments ± SD).

C  Ceramide and sphingomyelin levels in MAM isolated from WT and PS-DKO MEFs. Lipid units are represented as nmol/μg of protein over total nmoles of lipids analyzed (average of *n* = 3 independent experiments ± SD).

D  *De novo* synthesis of ceramide (Cer) and sphingomyelin (SM) in WT and PS-DKO MEFs (average of *n* > 5 independent experiments ± SD).

E  Activities of acid (aSMase) and neutral (nSMase) sphingomyelinases before and after BI (average of *n* = 5 independent experiments ± SD).

Data information: *$P < 0.05$. Analysis by unpaired *t*-test.

connections could explain the increased ceramide in mitochondrial membranes in AD.

To address this, we analyzed ceramide synthesis and SMase activity *in vitro*, using subcellular fractions from WT and PS-DKO cells. The results indicate that MAM indeed participates in regulating sphingolipid metabolism (Fig 5A and Appendix Fig S5A). Moreover, SMase activities were upregulated significantly in subcellular fractions from PS-DKO cells compared to controls (Fig 5A), as well as in MAM from PS1-KI$^{M146V}$ mouse brain (Appendix Fig S5B). In agreement with these results, Western blot analysis revealed a remarkable increase in the localization of nSMase to MAM in mutant cells compared to WT (Fig 5B), suggesting higher recruitment of SMase to these ER–mitochondria contacts.

To explore this further, we incubated PS-DKO and control cells with fluorescent sphingomyelin and analyzed its localization and conversion to ceramide in mitochondrial membranes by thin-layer chromatography (TLC). Presenilin-mutant cells showed a substantial decrease in fluorescent sphingomyelin intensity (Fig 5C) which was

paralleled by an increase in fluorescent ceramide (Fig 5C), implying that upregulated SMase activity may be responsible for this inverse behavior. Remarkably, the elevated deposition of ceramide at mitochondria disappeared when mutant cells were treated with BI (Fig 5C). These data suggest that the effect of BACE1 inhibition in enhancing mitochondrial respiration (Fig 1E and Appendix Fig S1N) may occur via the attenuation of sphingolipid metabolism in mutant cells.

Why is there an increased recruitment of nSMase to these ER regions in PS-mutant cells? It is well known that SMase activity is modulated by membrane characteristics and lipid composition (De Tullio *et al*, 2007). Notably, SMase activity is higher in lipid raft-like domains, such as MAM, where liquid-ordered and liquid-disordered phases coexist (Silva *et al*, 2009). In addition, nSMase shows increased affinity for membranes enriched in anionic phospholipids (Wu *et al*, 2011). In particular, the activity of nSMase2 is stimulated upon its binding to phosphatidylserine (PtdSer) (Wu *et al*, 2011). To see whether elevated PtdSer might be behind the increased recruitment of nSMase activity to MAM in mutant cells, we analyzed

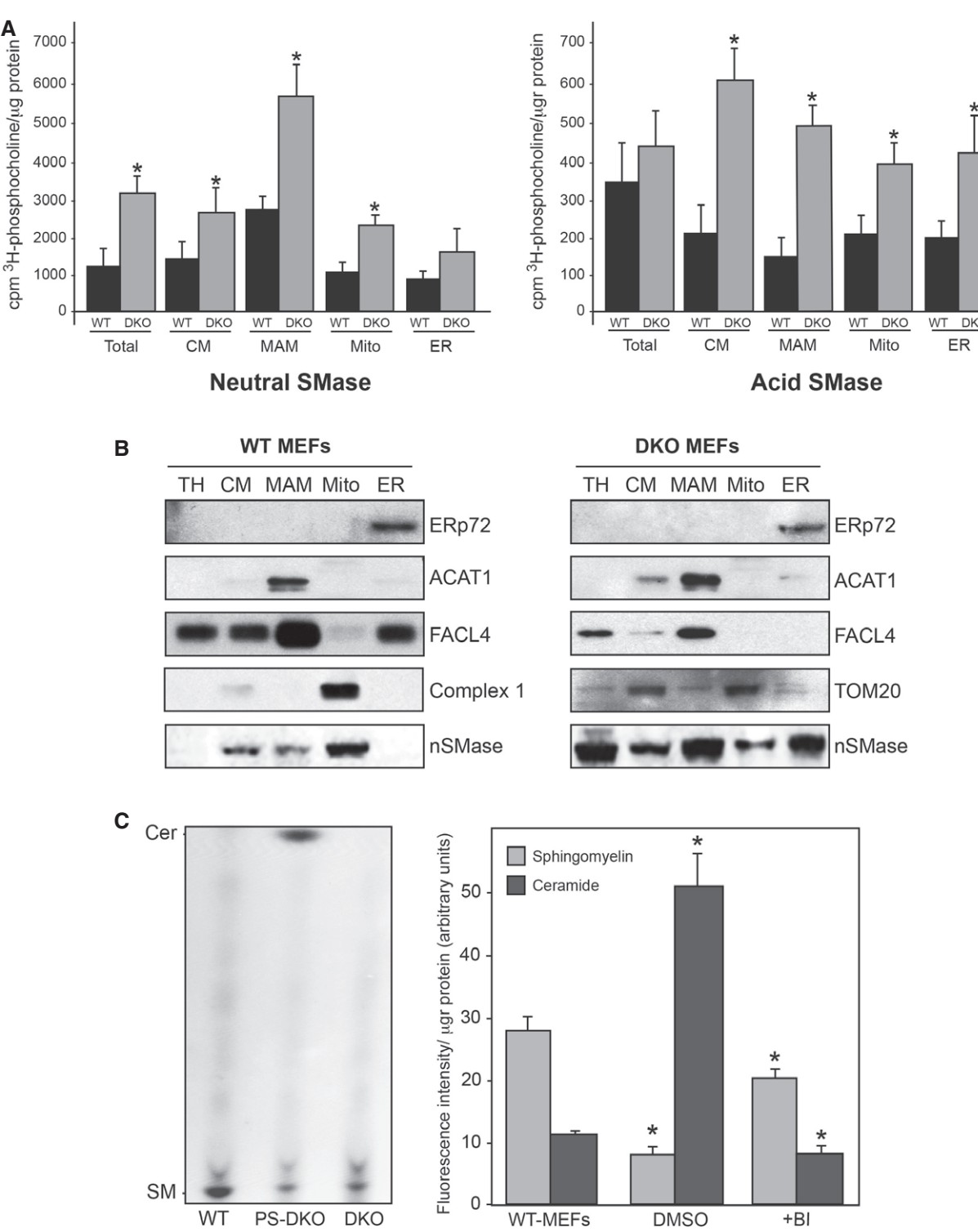

**Figure 5. MAM participates in the regulation of sphingolipid metabolism.**

A Distribution of SMase activity in the indicated subcellular compartments in WT and PS-DKO MEFs. Note the overall increase in nSMase levels in the PS-DKO cells (average of $n = 5$ independent experiments $\pm$ SD). *$P < 0.05$. Analysis by unpaired $t$-test.

B Western blot of the indicated fractions from WT and PS-DKO MEFs (probed with antibodies against the indicated markers) to detect the levels of nSMase protein in the indicated compartments (30 μg of protein per lane). TH, total homogenate; CM, crude membrane fraction.

C Sphingolipid levels in mitochondrial membranes extracted from the indicated cells after addition of fluorescent sphingomyelin (left, detection in TLC plates; right, quantitation) (average of $n = 5$ independent experiments $\pm$ SD). *$P < 0.05$ vs. WT levels. Analysis by unpaired $t$-test.

the content of PtdSer in WT and PS-DKO homogenates and in subcellular fractions. In agreement with our hypothesis, we found a significant increase in the amount of PtdSer in PS-DKO membranes, which was most pronounced in the MAM domains of the ER (Appendix Fig S5C). This result also supports the idea that increases in PtdSer due to C99-mediated upregulation of MAM are driving the increased SMase activity in these ER regions. Consistent with this proposed mechanism, inhibition of C99 production (with BI) reduced the PtdSer content of PS-DKO membranes to control levels (Appendix Fig S5D), while at the same time reversing the alterations in both sphingomyelin (Appendix Fig S5E) and ceramide (Appendix Fig S5F).

Taken together, we conclude that retention of uncleaved C99 in MAM in presenilin-deficient cells upregulates both the synthesis and catabolism of sphingomyelin in these regions of the ER, likely accounting for the increased ceramide in mitochondrial membranes (Fig 4A) via ER–mitochondrial connections.

### Mitochondrial dysfunction in AD is caused by upregulated sphingolipid turnover

The detrimental effects of ceramide on mitochondrial functionality have been shown extensively (Kogot-Levin & Saada, 2014). Thus, we speculated that the upregulation of SM turnover at MAM and the subsequent local increase in ceramide could be the underlying cause of the respiratory deficits seen in AD (Du *et al*, 2010; Swerdlow *et al*, 2014). To test this idea, we measured respiration in PS-DKO mutant cells treated with 5 μM myriocin, a specific inhibitor of serine palmitoyltransferase, the first step in the *de novo* pathway to synthesize sphingolipids, including ceramide. Inhibition of sphingolipid synthesis by myriocin resulted in slight decreases in sphingomyelin (Appendix Fig S6A) but in significant reductions in ceramide content (Appendix Fig S6B) in our cell models, with changes in the latter more pronounced in PS-DKO cells (Appendix Fig S6B). In agreement with our hypothesis, this reduction in ceramide rescued the bioenergetic defect in these mutant cells (Fig 6A).

Ceramide has been shown to provoke changes in mitochondrial lipid composition, altering its membrane potential and permeability (Kogot-Levin & Saada, 2014). Notably, the lipid composition of mitochondrial membranes is crucial for the stabilization and assembly of mitochondrial respiratory complexes into supercomplexes (also called respirasomes) necessary for optimal respiratory chain function (Acin-Perez & Enriquez, 2014). Therefore, it is possible that ceramide interferes with bioenergetics by destabilizing or preventing supercomplex assembly. To assess this, we used blue-native gel electrophoresis (Acin-Perez *et al*, 2008) to examine the activity (Fig 6B and C) and assembly status (Appendix Fig S6C and D) of supercomplexes in mitochondria from WT and PS-DKO cells, from DAPT-treated WT cells, and from PS-DKO cells incubated with BI and myriocin (Fig 6B). Measurements by in-gel staining of the activities of respiratory chain complexes I and IV (Fig 6B and C) and Western blotting to detect subunits of complexes I and III from PS-DKO and DAPT-treated WT cells (Appendix Fig S6C and D) showed a decrease in the activity of supercomplexes I+III+IV, I+III, and III+IV, which could be rescued after treatment with BI and myriocin (Fig 6D and E). Importantly, these changes in supercomplex activities and assembly were not due to alterations in the expression of individual complex subunits (Appendix Fig S6E).

To corroborate these results *in vivo*, we analyzed mitochondrial respiration and supercomplex activity in mitochondria isolated from brain tissue from PS1-KI$^{M146V}$ mice at various ages (Appendix Fig S7A and B). Interestingly, while MAM defects were already present in fetal cortical neurons (Appendix Fig S3A), decreases in mitochondrial respiration became significant only after 3 months of age (Appendix Fig S7A). In agreement with our previous results, this bioenergetic defect correlated with a significant decrease in supercomplex activity in mutant samples compared to controls (Appendix Fig S7C and D).

Taken together, these results indicate that the bioenergetic defects in AD are likely to be the consequence of upregulated sphingolipid turnover and increased ceramide content in mitochondria, triggered by the retention of C99 at the MAM. This elevation in ceramide levels alters mitochondrial membrane properties, likely hindering the assembly and activity of respiratory supercomplexes. Moreover, these data suggest that while mitochondrial dysfunction is an early and significant defect in AD, it is not a primary insult in the pathogenesis of the disease, but rather is a consequence of MAM dysfunction.

## Discussion

In previous reports, we showed that γ-secretase activity is localized in MAM (Area-Gomez *et al*, 2009) and that alterations in γ-secretase activity result in the upregulation of MAM function and in increased ER–mitochondria apposition (Area-Gomez *et al*, 2012). We now show that the γ-secretase substrate C99, in addition to its endosomal localization, is also present in MAM domains. Thus, both the γ-secretase enzyme activity (i.e., presenilins) and its direct substrate (i.e., C99) are located in the same compartment, where the former can cleave the latter. Moreover, chemical and genetic alterations of γ-secretase activity provoke a significant increase in the amount of this APP processing fragment in ER-MAM regions. The increased presence of C99 in MAM causes the upregulation of MAM functionality (as measured by ACAT1 activity) and greater apposition between ER and mitochondria. In addition, the higher concentration of MAM-localized C99 induces the recruitment of sphingomyelinase to this ER domain and the subsequent deregulation of sphingolipid homeostasis, followed by mitochondrial dysfunction.

These results support a model in which, in addition to Aβ, increased C99 plays an early role in AD pathogenesis, via altered MAM function. Of course, our results do not exclude the possibility that C99 has other roles in the pathogenesis of AD. In fact, increases in C99 were already shown to contribute to other aspects of the pathogenesis of the disease (Saito *et al*, 2011; Lauritzen *et al*, 2012), including endosomal dysfunction (Jiang *et al*, 2010), hippocampal degeneration (Lauritzen *et al*, 2012), and altered Tau proteostasis (Moore *et al*, 2015). In addition, elevations in C99 are toxic to neurons (Neve *et al*, 1996), correlating with symptoms of the disease (Rockenstein *et al*, 2005; Tamayev *et al*, 2012). Importantly, we note that although much of our data were obtained using FAD models and cells from FAD patients containing mutations in presenilins, alterations in γ-secretase activity and increased levels of C99 have been detected in sporadic AD patients as well (Fukumoto *et al*, 2002; Yang *et al*, 2003; Li *et al*, 2004; Pera *et al*, 2013).

We propose that, while the majority of C99 resides in endosomes, C99 can traffic to MAM regions in the ER, where it is cleaved rapidly

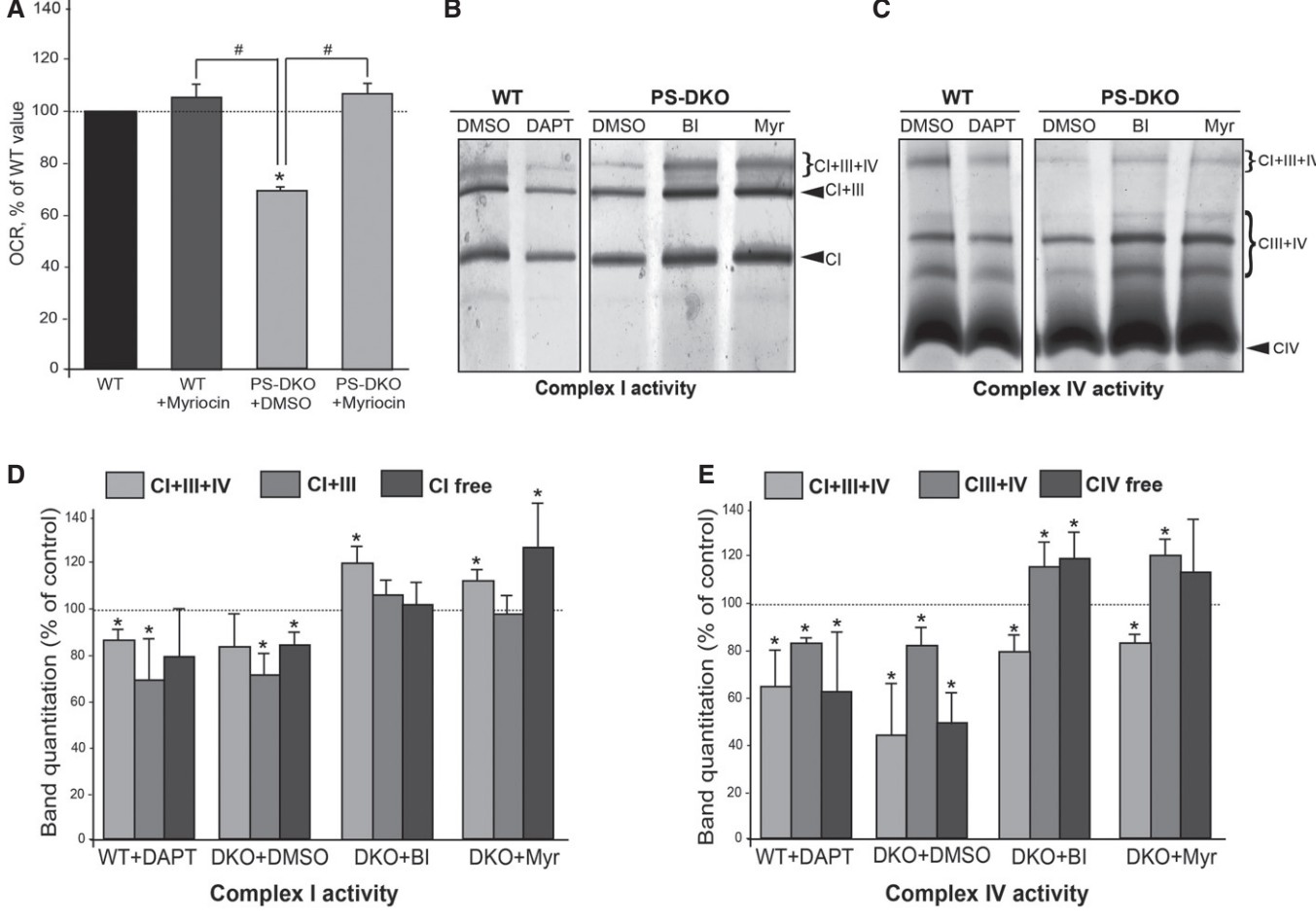

**Figure 6. Mitochondrial dysfunction is the consequence of increased sphingolipid turnover.**

A   Respiratory chain deficiency [as measured by initial oxygen consumption rate (OCR)] in PS-DKO cells was rescued after treatment with myriocin (inhibitor of the *de novo* sphingolipid synthesis pathway) (average of $n = 3$ independent experiments $\pm$ SD). *$P < 0.05$, compared to WT; #$P < 0.05$, comparisons indicated on graph. Analysis by unpaired *t*-test.

B, C   In-gel complex I (B) and complex IV (C) activity staining in mitochondria from WT and PS-DKO cells after the indicated treatments.

D, E   Quantification of specific bands shown in (B) and (C). Note that chemical or genetic inhibition of γ-secretase results in decreased supercomplex I+III+IV activity. This effect can be rescued by inhibition of C99 production [with a BACE1 inhibitor (BI)] or by inhibition of ceramide production with myriocin (Myr). Dotted lines denote baseline levels (average of $n = 3$ independent experiments $\pm$ SD). *$P < 0.05$ vs. baseline levels. Analysis by unpaired *t*-test.

Source data are available online for this figure.

by γ-secretase to produce Aβ and AICD (Area-Gomez *et al*, 2009; Schreiner *et al*, 2015). We note that although the mechanism by which C99 translocates to ER-MAM is unknown, recent work has demonstrated the existence of ER-endosome contacts (Rowland *et al*, 2014) where lipid and protein exchange may occur (Wilhelm *et al*, 2017). Thus, it is possible that C99 is delivered to ER via a similar mechanism, where it regulates the interaction between ER and mitochondria. Furthermore, the localization of C99 at MAM clarifies why C99, an ER-localized protein (Matsumura *et al*, 2014), was also detected in mitochondria (Devi & Ohno, 2012). Similarly, since C99 processing occurs at MAM, this could explain why Aβ has been found to colocalize with mitochondria (Hansson Petersen *et al*, 2008; Xie *et al*, 2013).

We show here that chemical or genetic alteration of γ-secretase activity results in an increase in unprocessed C99 in MAM. This, in turn, provokes the recruitment of SMase activity to MAM and the upregulation of sphingolipid turnover at these sites. These findings have mechanistic implications. Specifically, one of the MAM functions that is increased in AD is the synthesis of PtdSer by phosphatidylserine synthases 1 and 2 (PTDSS1/2) (Vance, 2014; Kannan *et al*, 2017 #2382, Wu & Voelker, 2004; Area-Gomez *et al*, 2012 #234), resulting in higher levels of this phospholipid in MAM and other membranes in AD cells and tissues. Considering the affinity of nSMase for PtdSer (Wu *et al*, 2011), we believe that the activation of nSMase by C99 is due, at least in part, to the upregulation of PTDSS1/2 in MAM, triggered by the increased concentration of this APP fragment. The details of this mechanism will require further investigation.

These results also help clarify a number of previous observations. First, as MAM contains γ-secretase and SMase activities, our data help explain why changes in APP processing can induce

alterations in sphingolipid regulation (Filippov *et al*, 2012; Lee *et al*, 2014). Second, upregulation of SMase and the resulting increase in ceramide are known to alter the size and composition of lipid raft domains (Dinkla *et al*, 2012), such as MAM. Therefore, an increased localization of C99 in MAM in AD may explain the upregulation of ER–mitochondria connections and MAM functionality seen in the disease (Area-Gomez *et al*, 2012; Hedskog *et al*, 2013). Finally, given the detrimental effect of ceramide on mitochondrial super-complex assembly and respiratory chain activity (Zigdon *et al*, 2013), we conclude that its accumulation is likely to be a primary cause of mitochondrial dysfunction in AD.

This last conclusion disagrees with proposals that the accumulation of $A\beta_{42}$ oligomers in mitochondria triggers the mitochondrial defects seen in AD (Manczak *et al*, 2006). Rather, our results show that PS-DKO and DAPT-treated cells, in which $A\beta$ production is inhibited, can nevertheless recapitulate the mitochondrial deficits seen in AD. This finding suggests that mitochondrial deficiencies are due to increased levels of C99 rather than to elevated production of longer $A\beta$ species, since bioenergetic deficiency can occur in the absence of $A\beta$. We believe that the discrepancy between our results and those of others showing reductions in mitochondrial respiration after incubation with $A\beta$ (Casley *et al*, 2002) is due mainly to the use of unphysiologically high concentrations of this peptide (Casley *et al*, 2002). Thus, we propose that MAM and mitochondrial alterations are caused by an increased ratio of C99:A$\beta$, rather than by an increased ratio of $A\beta_{42}$:$A\beta_{40}$. In agreement with this, the increase of C99 in mitochondria in AD has been described before, correlating with mitochondrial respiratory defects that could be rescued by partial deletion of *BACE1* (Devi & Ohno, 2012). Finally, our results linking C99, rather than higher levels of $A\beta_{42}$, to mitochondrial dysfunction help explain how mitochondrial alterations can occur early in AD pathogenesis (Balietti *et al*, 2013), preceding the appearance of $A\beta$-containing plaques (Yao *et al*, 2009).

In summary, our data demonstrate that increased levels of unprocessed MAM-localized C99 are a driver of mitochondrial dysfunction in AD, mediated by the loss of sphingolipid homeostasis at ER–mitochondria connections. Equally important, while the toxicity of $A\beta$ is undeniable, we suggest a role for elevated C99 (Rockenstein *et al*, 2005) and MAM deregulation (Schon & Area-Gomez, 2010, 2013) in the pathogenesis of the disease, thus providing a new framework for understanding the link between alterations in APP processing and lipid homeostasis as seminal effectors of AD pathogenesis. Further work is required to fully elucidate the mechanism by which MAM-C99 induces these alterations and validate its relevance in the pathogenesis of this devastating disease.

## Materials and Methods

### Cells, animals, and reagents

AD and control cell lines were obtained from the Coriell Institute for Medical Research (Camden, NJ, USA). SH-SY5Y and COS-7 cells were obtained from the American Type Culture Collection. Other PS1-mutant FAD cells were the kind gift of Dr. Gary E. Gibson (Cornell University). WT, PS1-KO, PS2-KO, and PS1/2-DKO (called PS-DKO) mouse MEFs were provided by Dr. Bart De Strooper (University of Leuven). APP/APLP2-KO (called APP-DKO) (Herms

*et al*, 2004) and PS1-KI$^{M146V}$ knock-in mice (Guo *et al*, 1999). All experiments were performed according to a protocol approved by the Institutional Animal Care and Use Committee of the Columbia University Medical Center and were consistent with the National Institutes of Health Guide for the Care and Use of Laboratory Animals. Mice were housed and bred according to international standard conditions, with a 12-h light/12-h dark cycle, and sacrificed at 3, 5, 7, 8, and 12 months of age. Brains were removed and homogenized for Western blot and Seahorse analysis. All the experiments were performed on at least three mice per group.

We used antibodies to ACAT1 (Abcam, ab39327), APP C-terminal (Sigma; A8717, polyclonal), APP-C99 [Covance; SIG-39320-200 (6E10), monoclonal], the α-subunit of mitochondrial ATP synthase (complex V) (Invitrogen; 459240), the α-subunit of ATPase (Abcam, ab7671), BACE1 (Cell Signaling; D10E5), CANX (Chemicon, MAB3126), CDH2 (ref), complex I subunit NDUFA9 (Abcam; ab14713), complex III subunit core-1-ubiquinol-cytochrome *c* reductase (Abcam; ab110252), OxPhos complex IV subunit IV (COX IV) (Abcam; ab14744), Ergic53/p58 (Sigma; E1031), Erlin-2 (Cell Signaling; #2959), ERp72 (Cell Signaling, D70D12), FACL4 (Abgent, AP2536b), GM130 (BD Transduction Laboratories, 610822), G6PC (ref), Lamp2 (Novus biologicals; NBP1-71692), Na$^+$/K$^+$ ATPase (Abcam, ab7671), PEMT (a gift of Jean Vance, University of Alberta), Presenilin 1 (Calbiochem; PC267; NOVUS biologicals; EP1998Y), Rab5a (NOVUS Biologicals; NBP1-58880), Rab7a (Novus Biologicals; NBP1-87174), TRAP-α (ref), nSMase (Thermo Scientific; PA5-24614), total OXPHOS mouse cocktail (abcam, ab110413), TOM20 (Santa Cruz; sc-11415), β-tubulin (Sigma; T4026), vinculin (Sigma, V4505), and VDAC1 (Abcam; 34726). TLC silica plates were from EMD Biosciences (5748-7). Ceramide (22244), sphingomyelin (S0756), cholesteryl palmitate (C6072), cholesteryl oleate (C9253), lipid markers for TLC (P3817), α-secretase inhibitor TAPI-1 (SML0739), cytochrome *c* from horse heart (C2506), 3,3′-diaminobenzidine tetrahydrochloride hydrate (D5637), GI254023X (SML0789), β-secretase inhibitor IV (Calbiochem; 565788), γ-secretase inhibitor DAPT (D5942), antimycin A (A8674), FCCP (carbonylcyanide p-(trifluoromethoxy)phenylhydrazone) (C2920), NADH Grade II, disodium salt (Roche; 10128023001), nitro blue tetrazolium (N5514-25TA1), oligomycin (O4876), rotenone (R8875), imatinib mesylate (Gleevec®, SML1027), and serine palmitoyltransferase inhibitor myriocin (M1177) were from Sigma. Fluorescent lipids BODIPY-FL C6 ceramide complexed to BSA (N22651) and BODIPY-FL C12-sphingomyelin (D7711) were from Invitrogen. Radiolabelled $^3$H-serine and $^3$H-cholesterol were from Perkin Elmer; fatty acid-free bovine serum albumin (FAF-BSA) was from MP Biomedical (820472). Amyloid β peptides 40 aa and 42 aa were from Biopolymer Laboratory (UCLA), and AICD peptide was from Genescript Corporation (Piscataway, NJ, USA).

### Seahorse analysis

Respirometry of cultured cells was performed using the XF24e Extracellular Flux Analyzer (Seahorse Bioscience). Oxygen consumption was measured in basal conditions (Seahorse media with 25 mM glucose and 2 mM pyruvate) and after the sequential addition of 1 μM oligomycin (complex V inhibitor), 0.75 μM FCCP (uncoupler), and 1 μM rotenone/1 μM antimycin A (complex I and complex III inhibitors, respectively). All results were averages of five or more

biological replicates. Every biological replicate consisted of three technical replicates. For every technical replicate, we plated equal number of cells (25,000 cell/well when MEFs were used, and 50,000 cells/well when human primary fibroblasts were analyzed). The number of cells was also counted after every respirometry assay to correct for cell death. All oxygen consumption (OCR) data were normalized by the number of viable cells or by protein quantity when isolated mitochondria were used.

For permeabilization assays, the cell culture medium was replaced by the mitochondrial assay solution (70 mM sucrose, 220 mM mannitol, 5 mM $KH_2PO_4$, 5 mM $MgCl_2$, 2 mM HEPES, 1 mM EGTA and 0.2% FAF-BSA, pH 7.4) containing 10 nM of the XF Plasma membrane permeabilizer reagent XF PMP (Seahorse Bioscience #102504-100) and pyruvate/malate (for complex I assays) or succinate/rotenone (for complex II assays). Oxygen consumption was measured at States 2, 3, 4, and uncoupling after sequential addition of 3 mM ADP, 4 μM oligomycin, 6 μM FCCP, and 4.5 μM Antimycin A.

To analyze mitochondrial respiration in mouse tissues, mitochondria were isolated from WT and PS1-KI$^{M146V}$ mouse brain. Mouse brains were homogenized in ~10 volumes of homogenization buffer (210 mM mannitol, 70 mM sucrose, 5 mM HEPES, and 1 mM EGTA) and then centrifuged at 900 × *g* for 10 min at 4°C. The remaining supernatant was centrifuged at 9,000 × *g* for 10 min at 4°C, and the resulting pellets were resuspended in washing buffer (210 mM mannitol, 70 mM sucrose, 5 mM HEPES, 1 mM EGTA, and 0.5% FAF-BSA pH 7.2) and centrifuged again at 8,000 × *g* for 10 min at 4°C. The pellets, containing mitochondria, were resuspended in mitochondrial assay solution, and protein was quantitated using the BCA Protein Assay kit (Thermo Scientific #23227). For complex I experiments, 8 μg of protein was added to each well and for complex II analysis 6 μg per well. Analyses in the Seahorse analyzer were performed as described in the permeabilization assays.

## Culture of primary mouse cortical neurons

Cortexes from four 14-day-old embryos were cut in pieces and washed in 45% glucose in PBS. After that, brain tissues were resuspended in 1 ml trypsin diluted in 45% glucose in PBS (1:1 v/v) and incubated at 37°C for 20 min. Samples were added to 500 μl horse serum and 10 units of DNase and incubated for 10 min at room temperature until debris sank to the bottom of the tubes. The non-debris fraction was pelleted at 800 × *g* for 10 min and resuspended in Neurobasal Medium (Life Technologies; 21103-049) supplemented with 200 mM glutamine. Cells were counted and seeded on coverslips coated with poly-ornithine and laminin.

## Plasmid constructs and transfections

Plasmids were constructed using standard techniques. In brief, APP fragments AICD and C99 were amplified from pCAX APP-695 (Young-Pearse *et al*, 2007), using forward primer 5′-cccgctagc<u>ctcgag</u>**ATG**CTGAAGAAGAAACAGTACACATCCATTC-3′ for AICD, and 5′-ccc<u>ggatcc</u>**ATG**GATGCAGAATTCCGACATGACTC-3′ for C99, with a single reverse primer 5′-ccc<u>ggatcc</u>aagctt**CTA**GTTCTGCATCTGCTCAAAGAACTTG-3′ for both; restriction sites for subcloning are

underlined and the start/stop codons are in bold. The PCR products were cut with *Xho*I+*Bam*HI (for AICD) or with *Bam*HI (for C99) and subcloned into the corresponding sites in pGFP-N3 (Clontech). Plasmid C99-GFP was kind gift of Dr. Albert Lleo. For the construction of this plasmid, C99 amplified from APP770 GFP using forward primer_(Scott): (HindIII) 5′ gc<u>aagcttg</u>cagaattccgacatgactcagga 3′ and reverse primer cw mini LRP 3′ (psectag 2B mini LRP GFP primers). The fragment was subcloned into psectag 2B with HindII/NotI. All plasmids were verified by restriction analysis and sequencing. Cells were transfected using Lipofectamine™ Transfection Reagent (Thermo Fisher Scientific, Life Technologies) according to the manufacturer's instructions.

## Subcellular fractionation and Western blotting

Purification of ER, MAM, and mitochondria was performed and analyzed as described (Area-Gomez *et al*, 2009). For C99 detection, samples were run in 4–12% Bis–Tris gels (Criterion XT Precast Midi Gels, BioRad) in XT MES buffer.

## Electron microscopy analysis

Samples were fixed in 2.5% glutaraldehyde in 0.1 M sodium cacodylate buffer, enrobed in 4% gelatin, postfixed with 1% osmium tetroxide (aq) followed by 2% uranyl acetate, dehydrated through a graded series of ethanol, and embedded in LX112 resin (LADD Research Industries, Burlington, VT, USA). Ultrathin sections were cut onto nickel grids with a Leica Ultracut UCT (Leica Microsystems, Wetzlar, Germany).

## Antigen retrieval immunolabeling

Sections were etched with saturated sodium metaperiodate for 1 h, washed with PBS, blocked with 1% BSA, and incubated with primary antibody overnight at 4°. The next day, they were washed and then incubated in 6 nm goat anti-rabbit gold (Aurion, NL), for 2 h at room temperature. Sections were counterstained with uranyl acetate and viewed on a JEOL JEM-1400Plus transmission electron microscope at 120 kV.

## Inhibition of α-, β-, and γ-secretase activity

To inhibit γ-secretase activity, cells were treated with 10 μM DAPT, a highly specific inhibitor of this enzyme complex. For β-secretase inhibition, cells were treated with 100 nM β-secretase inhibitor IV (BI) or different doses of imatinib lysate (Gleevec). To inhibit α-secretase, cells were treated with 5 μM of TAPI-1 (Enzo Life Sciences). Inhibition of aSMase and the nSMase activities was performed using 10 μM desipramine or 5 μM GW4869, respectively. To inhibit serine palmitoyltransferase activity, the cells were treated with 5 μM myriocin. Incubations with all drugs were for 12–16 h.

## Staining of lipid droplets

Staining of lipid droplets was performed using HCS LipidTox™ Deep Green neutral lipid stain (Invitrogen H34475) according to the manufacturer's instructions. Lipid droplet staining was quantified

using ImageJ. The different values represent the product of the intensity and the area covered by the fluorescent signal above background in every cell examined.

## Sphingolipid synthesis in cultured cells

Cells were incubated for 2 h with serum-free medium to ensure removal of exogenous lipids. The medium was then replaced with MEM containing 2.5 μCi/ml of $^3$H-serine for the indicated periods of time. The cells were washed and collected in PBS, pelleted at 2,500 × $g$ for 5 min at 4°C, and resuspended in 0.5 ml water, removing a small aliquot for protein quantification. Lipid extraction was done in three volumes of chloroform:methanol:HCl (2:1:0.5 v/v/v) added to the samples. Samples were vortexed and centrifuged at 8,000 × $g$ for 5 min; the organic phase was blown and dried under nitrogen. Dried lipids were resuspended in 30 μl of chloroform:methanol (2:1 v/v) and applied to a TLC plate. Sphingolipids were separated using a solvent composed of chloroform/methanol/0.22% CaCl$_2$ (60:35:8 v/v/v). Development was performed by exposure of the plate to iodine vapor. The spots corresponding to the relevant sphingolipids (identified using co-migrating standards) were scraped and counted in a scintillation counter (Packard Tri-Carb 2900TR).

## Lipidomic analyses

Lipids were extracted from equal amounts of material (30 μg protein/sample). Lipid extracts were prepared via chloroform–methanol extraction, spiked with appropriate internal standards, and analyzed using a 6490 Triple Quadrupole LC/MS system (Agilent Technologies, Santa Clara, CA) as described previously (Chan *et al*, 2012). Glycerophospholipids and sphingolipids were separated with normal-phase HPLC using an Agilent Zorbax Rx-Sil column (inner diameter 2.1 × 100 mm) under the following conditions: mobile phase A (chloroform:methanol:1 M ammonium hydroxide, 89.9:10:0.1, v/v/v) and mobile phase B (chloroform:methanol:water: ammonium hydroxide, 55:39.9:5:0.1, v/v/v/v); 95% A for 2 min, linear gradient to 30% A over 18 min and held for 3 min, and linear gradient to 95% A over 2 min and held for 6 min. Quantification of lipid species was accomplished using multiple reaction monitoring (MRM) transitions that were developed in earlier studies (Chan *et al*, 2012) in conjunction with referencing of appropriate internal standards: ceramide d18:1/17:0 and sphingomyelin d18:1/12:0 (Avanti Polar Lipids, Alabaster, AL, USA). Values are represented as mole fraction with respect to total lipid (% molarity). For this, lipid mass (in moles) of any specific lipid is normalized by the total mass (in moles) of all the lipids measured (Chan *et al*, 2012). In addition, all of our results were further normalized by protein content.

## Analysis of sphingolipid synthesis in subcellular fractions

Cellular fractions were isolated from MEFs as described (Area-Gomez *et al*, 2009). Two hundred micrograms was incubated in a final volume of 200 μl of 100 mM HEPES pH 7.4, 5 mM DTT, 10 mM EDTA, 50 μM pyridoxal phosphate, 0.15 mM palmitoyl-CoA, and 3 μCi/ml $^3$H-Ser for 20 min at 37°C. The reaction was stopped by addition of three volumes of chloroform/methanol (2:1

v/v). Lipid extraction and TLC analysis were performed as described above.

## Analysis of sphingomyelinase activity

One hundred micrograms of protein was assayed in 100 mM of the appropriate buffer (Tris/glycine for pH 7.0–9.0 and sodium acetate for pH 4.0–5.0), 1.55 mM Triton X-100, 0.025% BSA, 1 mM MgCl$_2$, and 400 μM bovine brain sphingomyelin spiked with 22,000 dpm of [$^3$H]-bovine sphingomyelin (1 nCi/sample). Reactions were carried out in borosilicate glass culture tubes at 37°C, overnight, followed by quenching with 1.2 ml of ice-cold 10% trichloroacetic acid, incubation at 4°C for 30 min, and centrifugation at 2,000 rpm at 4°C for 20 min. One milliliter of supernatant was transferred to clean tubes, 1 ml of ether was added, the mixture vortexed, and centrifuged at 2,000 rpm for 5 min. Eight hundred microliters of the bottom phase was transferred to scintillation vials, 5 ml of Scintiverse BD (Fisher Scientific, Fair Lawn, NJ, USA) was added, and samples were counted.

## ACAT activity assay

To measure cholesterol esterification *in vivo*, cultured cells were incubated in serum-free medium for 2 h to remove all exogenous lipids. After that, 2.5 μCi/ml of $^3$H-cholesterol was added to FBS-free DMEM containing 2% FAF-BSA, allowed to equilibrate for at least 30 min at 37°C, and the radiolabeled medium was added to the cells for the indicated periods of time. Cells were then washed and collected in DPBS, removing a small aliquot for protein quantification. Lipids were extracted in three volumes of chloroform: methanol (2:1 v/v). After vortexing and centrifugation at 8,000 × $g$ for 5 min, the organic phase was blown to dryness under nitrogen. Dried lipids were resuspended in 30 μl of chloroform:methanol (2:1 v/v) and applied to a TLC plate along with unlabeled standards. A mixture of hexanes/diethyl ether/acetic acid (80:20:1 v/v/v) was used as solvent. Iodine-stained bands corresponding to cholesterol and cholesteryl esters were scraped and counted.

## Analysis of ER–mitochondrial apposition

Cells under were co-transfected with GFP-Sec61-β (Addgene plasmid #15108) and DsRed2-Mito (Clontech, #632421) at a 1:1 ratio, using Lipofectamine 2000 (Invitrogen, #11668-027) in serum-free DMEM. Twelve hours post-transfection, cells were analyzed as described (Guardia-Laguarta *et al*, 2014).

## Preparation of synthetic Aβ in different states of aggregation and Aβ$_{40}$/Aβ$_{42}$ detection

Lyophilized Aβ$_{40}$ and Aβ$_{42}$ peptides (American Peptide; 62-0-80; UCLA) were equilibrated at room temperature for 30 min and then resuspended in hexafluoro-2-propanol (HIFP) (Sigma; H8508) to 1 mM using a glass-tight Hamilton syringe with Teflon plunger. HIFP was allowed to evaporate in a fume hood and dried under vacuum in a SpeedVac (Savant Instruments) and kept at −20°C. Immediately prior to use, an aliquot was resuspended to 5 mM in DMSO followed by bath sonication for 10 min.

          

To analyze the effect of Aβ addition, a mix of $A\beta_{40}/A\beta_{42}$ at a ratio 10:1 was added to the cultured cells to a final concentration of 6,000 pg/ml for 24 h. For $A\beta_{42}$ oligomer formation, 5 mM of $A\beta_{42}$ in DMSO was diluted to 100 μM in ice-cold media, vortexed for 30 s, and incubated at 4°C for 24 h. $A\beta_{42}$ oligomers were added to the cultured cells to a final concentration of 5 or 10 μM for 24 h.

Detection of $A\beta_{40}/A\beta_{42}$ levels in cell media was performed by ELISA following manufacturer's instructions (WAKO ELISA kit 294-64701 for $A\beta_{40}$ and 292-64501 for $A\beta_{42}$).

### Quantitative reverse transcription–polymerase chain reaction (qRT–PCR)

Total RNA was extracted from MEFs using TRIzol® Reagent (Invitrogen 15596-018) according to the manufacturer's instructions and was quantified by NanoDrop2000 (Thermo Scientific). One microgram of total RNA was used to obtain cDNA by RT–PCR using a High Capacity cDNA Reverse Transcription Kit (Applied Biosystems; PN 4368813, 4374966). Real-Time PCR was performed in triplicate in a StepOnePlus™ Real-Time PCR System (Applied Biosystems; 4376600). The expression of each gene under study was analyzed using specific predesigned TaqMan Probes (PGC-1α, ppargc1a Mm01208835_m1; aSMase, smpd1 Mm00488319_g1; nSMase, smpd3 Mm00491359_m1). The forward and reverse primers $(5'{\to}3')$ for *Cox1* quantification were, respectively: TGCTAGCCGCAGGCATTACT and CGGGATCAAAGAAAGTTGTGT TT. The expression of each gene under study was analyzed using specific predesigned TaqMan Probes and normalized against *Gapdh* expression (Applied Biosystems, 4352339E) as an internal standard.

### Supercomplex analysis

Analysis and quantification of mitochondrial respiratory complexes by Western blot and enzymatic in-gel activity were carried out as described (Acin-Perez *et al*, 2008).

### Statistical analyses

All averages are the result of three or more independent experiments carried out at different times with different sets of samples. Tests of significance employed Student's *t*-test at $P < 0.05$, unless indicated otherwise; all error bars in the figures are ± SD. For the determination of ER–mitochondrial apposition, all images were taken randomly from a set of multiple fields. The degree of colocalization was analyzed by ImageJ, and data were compared using Mander's coefficient.

**Expanded View** for this article is available online.

## Acknowledgements

We thank Drs. Orian Shirihai and Marc Liesa (UCLA) for assistance with the Seahorse measurements, Dr. Huaxi Xu (Sanford Burnham Institute) for the APP-DKO MEFs and Dr. Mark Mattson (NIH) for the PS1 knock-in mice, Drs. Arancio and Teich for the APP-KO mice tissues used in these studies, Dr. Hua Yang (Columbia University) for mouse husbandry, and Drs. Marc Tambini, Ira Tabas, and Serge Przedborski for helpful comments. This work was supported by the Fundación Alfonso Martín Escudero (to M.P.); the Alzheimer's Drug Discovery Foundation, the Ellison Medical Foundation, the Muscular Dystrophy Association, the U.S. Department of Defense (W911NF-12-1-9159 and W911F-15-1-0169), and the J. Willard and Alice S. Marriott Foundation (to E.A.S.); the U.S. National Institutes of Health (P01-HD080642 and P01-HD032062 to E.A.S.; NS071571 and HD071593 to M.F.M.; R01-NS056049 and P50-AG008702 to G.D.P.; 1S10OD016214-01A1 to G.S.P. and F.P.M, and K01-AG045335 to E.A.-G.), the Lucien Coté Early Investigator Award in Clinical Genetics from the Parkinson's Disease Foundation (PDF-CEI-1364 and PDF-CEI-1240) to C.G.-L., and National Defense Science and Engineering Graduate Fellowship (FA9550-11-C-0028) to R.R.A.

## Author contributions

MP, DL, JM, and RRA performed and interpreted most of the experiments. KRV provided laboratory support. CG-L performed the confocal analysis experiments. YX, RBC, and GDP performed most of the lipidomics analysis. GSP, FPM, and ZZF performed the electron microscopy studies. RA-P and JAE performed the supercomplexes analysis. MFM provided the PS1-KI$^{M146V}$ mice. EAS conceived the project, interpreted most of the experiments, and wrote the manuscript. EA-G conceived the project, designed, performed, and interpreted most of the experiments, and wrote the manuscript.

## Conflict of interest

The authors declare that they have no conflict of interest.

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
