## [Review Process File · The EMBO Journal]

Manuscript EMBO-2017-96797

Increased localization of APP-C99 in mitochondria-associated ER membranes causes mitochondrial dysfunction in Alzheimer disease

Marta Pera, Delfina Larrea, Cristina Guardia-Laguarta, Jorge Montesinos, Kevin R. Velasco, Rishi R. Agrawal, Yimeng Xu, Robin B. Chan, Gilbert Di Paolo, Mark F. Mehler, Geoffrey S. Perumal, Frank P. Macaluso, Zachary Z. Freyberg, Rebeca Acin-Perez, Jose Antonio Enriquez, Eric A. Schon & Estela Area-Gomez

Corresponding author: Estela Area-Gomez, Columbia University Medical Center

Review timeline:

Submission date:	23 February 2017
Editorial Decision:	20 March 2017
Revision received:	24 July 2017
Editorial Decision:	14 August 2017
Revision received:	18 August 2017
Accepted:	01 September 2017

Editor: Andrea Leibfried

Transaction Report:

1st Editorial Decision

20 March 2017

Thank you for submitting your manuscript for consideration by the EMBO Journal. It has now been seen by three referees whose comments are shown below.

As you will see, the referees appreciate your findings. However, they also think that further insight is needed for publication in The EMBO Journal. Importantly,

- controls need to be added (referee #1, point 4; referee #2 points 1-2, 8)
- the physiological relevance of your conclusions needs to be better demonstrated (referee #2, points 3-4, 6)
- some observations need further explanations (referee #1, point 3; referee #3, points 2-3, 5)

Given the referees' positive recommendations, I would like to invite you to submit a revised version of the manuscript, addressing the comments of all three reviewers and especially those noted above. I should add that it is EMBO Journal policy to allow only a single round of revision, and acceptance of your manuscript will therefore depend on the completeness of your responses in this revised version. Please get in touch in case you would like to discuss individual revision points further.

Thank you for the opportunity to consider your work for publication. I look forward to your revision.

REFEREE REPORTS

Referee #1:

Summary

The authors of this manuscript have followed up on previous work (2009, 2012) in which they showed that gamma secretase activity is present in mitochondria-associated ER membranes (MAM) and that in models of Alzheimer disease alterations in the activity of this protein increases MAM function and the apposition between ER and mitochondria. They now report that the 99 aa C-terminal fragment (C99) of APP is present in MAM and that in models of AD the concentration of C99 in MAM is increased, resulting in increased sphingomyelin degradation. They conclude that mitochondrial function is thereby impaired, consistent with the mitochondrial defects observed in AD.

Major Comments

1. This study is, in general, well performed with many appropriate controls. Moreover, the topic of the study is very interesting and the results are novel. In addition, the manuscript is well written and well organized.

2. Lipid analyses: the way in which lipid amounts are reported - as "molar mass over total moles of lipids analyzed" (mol %; e.g. in Fig 4) - is somewhat problematic. It is not at all clear what these numbers represent. A much more definitive number would be to give the nmoles of each lipid/mg cell protein so that the reader can directly assess whether the amount of that lipid is increased or not under a specific condition. It is also very unclear what is meant by "total moles of lipids analyzed" (mol %). Importantly it is not stated which lipids were included in this total value? Please either quote the data as nmole lipid/mg protein or give the 100% value of total nmoles of lipids analyzed.

3. Sphingolipid metabolism: it is obviously complicated to analyze sphingolipid synthesis and degradation in this situation. For example in Fig 4D, the incorporation of [3H]serine into ceramide and sphingomyelin is higher in the DKO than in controls. However, this result does not necessarily mean that ceramide synthesis or SM synthesis is increased in the DKO. First, it is not possible from this experiment to determine if synthesis is increased or if degradation is decreased in the DKO. Second, SM is a precursor of ceramide (SM degradation), and ceramide is a precursor of SM (SM synthesis). If radiolabel in SM were derived from radiolabeled ceramide, and if the radiolabel (and therefore specific radioactivity) in ceramide were higher in DKO than in control, the labeling of SM would automatically be higher in the DKO without an increase in SM synthesis. This is not the only complicating scenario. Thus, the wording about increased sphingolipid synthesis needs to be carefully modified: there is in fact no evidence that the synthesis of ceramide or SM is increased in the DKO (see text page 8 etc). Nevertheless, the data on the SMase assays do indicate that SMase activity is higher in the DKO, and that the increase in ceramide is probably due to the increase in SMase activity rather than de novo ceramide synthesis.

4. Myriocin expts Fig 6A: as a positive control for these expts, it would be very appropriate to confirm that amounts of sphingolipids (e.g. ceramide, SM as nmol/mg protein) in MAM and mito are indeed reduced by myriocin under the conditions used in these experiments.

Minor Comments

1. Why not include Fig S2C as a main Fig rather than a Supp Fig?

2. Page 13, para 2. Further to the discussion on the role of phosphatidylserine the authors should consider adding some discussion of the role of the anionic lipid, PS, in mediating contacts between the ER and mitochondria [see Wu and Voelker (2004) JBC 279:6635 and a very recent paper from Prinz lab in J Lipid Res (2017)].

Referee #2:

Pera and colleagues provide an interesting study addressing the molecular mechanisms underlying neurotoxicity in Alzheimer's disease (AD). The authors suggest that increased localization of a specific APP fragment (C99) in mitochondria-associated ER membranes (MAM) causes mitochondrial dysfunction in AD. This conclusion is mostly based on the use of mouse embryonic fibroblasts (MEF) either lacking presenilin (PS) the catalytic subunit of gamma-secretase or overexpressing a mutant PS with reduced activity. With these cells the authors demonstrate altered lipid processing and composition of both MAMs and mitochondria. Finally, the authors demonstrate mitochondrial dysfunction in situations of increased C99 levels. This is a novel and beautiful cell biological and biochemical study addressing a central question in neurodegeneration research. However, my major criticism is two-fold. First, essential control experiments are missing. Second, the physiological relevance of the key findings is not provided as the study is mostly done with MEF cells.

The following points need to be addressed to improve the quality of the manuscript.

Major points:

1. The different MEF cell lines (WT, PS1 KO, PS2 KO, DKO) are not of the same origin and may show protein expression level changes that go well beyond the genetic differences (PS KO). For this reason it is state of the art to repeat at least the essential experiments with DKO cells re-transfected/reconstituted with either PS1 or PS2 or both to ensure the same background of the cells. This needs to be included.
2. BACE inhibitors are prone to off-target effects, e.g. on cathepsin D, which is a main reason for several failed clinical trials with BACE inhibitors. Thus, key experiments in the study need to be repeated with a knock-down or knock-out/CRISPR of BACE1 or at least a BACE inhibitor with a different chemical structure.
3. Please show lipid and mitochondrial changes in BACE1-deficient mice and in APP-deficient mice or in APP-transgenic mice. They do all have altered C99 levels compared to WT mice and thus should show changes similar to MEF cells, if the proposed function and mechanism are true. Importantly, show WT mice as a control. They are currently missing in figure S7.
4. Many experiments are done in PS DKO cells, which (artificially) increase C99 levels. PS has around 100 different substrates and has additional functions in calcium signaling. Thus, I am not yet convinced that the artificial situation of PS DKO is relevant to the situation in vivo.
5. In mice, so far only one PS mutant is tested. This represents one form of familial AD, which makes up about 1% of all AD cases. 99% of AD are sporadic and do not involve PS abnormalities and probably also not C99 abnormalities. Yet, the title of the manuscript is about mitochondrial dysfunction in general in AD. Thus, you either need to test another AD model as well or change the title to better reflect the content of the manuscript.

Minor points:

6. In the introduction the authors claim that increased C99 contributes to AD. This is an overselling of two previous publications and does not reflect the general state of the field.
7. Fig. 1C: show C99 and C83 levels. It is known that under conditions of gamma-secretase inhibition there is more conversion of C99 to C83.
8. Figure 2A: include an APP ko to ensure that the APP CTFs are specific bands. Also include - ideally in all gels - molecular weight markers. APP ko material will also help to ensure the specificity of the apparent AICD band in figure S2I. AICD is typically very difficult to detect.
9. Figure 2C: better separate the C99 and C83. Currently, it looks like one band instead of two. Indicate for the C99 gels which antibody was used. This is not clear from the methods section (N- or C-terminal antibody to C99).

Referee #3:

The authors have investigated the effects of the accumulation of the C99 fragment of APP in MAMs on mitochondrial respiratory chain function and on ceramide metabolism. They show significant decreases on oxygen consumption in cells from FAD patients, in MEFs in which both PS genes are knocked out, and in mitochondria from a mouse model of AD. They show that C99 accumulates in MAMs in the absence of gamma secretase activity and that this affects the turnover of ceramides, ultimately resulting in an increased ceramide content in mitochondria, that is associated with

decreased formation of respiratory chain supercomplexes, which they argue underlies the oxygen consumption defect.

This is a carefully executed study that I think goes some way to demonstrating that accumulation C99 at the MAM may underlie early AD pathology by interfering with mitochondrial respiratory chain activity.

I have the following comments:

- (1) It might be useful to show the Seahorse traces from Fig. 1 in the Supplemental data. Was the defect in CR compensated by an increase in ECAR? What about maximum uncoupled rate?
- (2) In Figure 2A why is Lamp 1 completely localized to the mito fraction and in 2B why is VDAC in every fraction of the gradient? Is that true for other mitochondrial markers? In 2 D what is the explanation for the fact that the C99 positive foci are in many cases much larger than mitochondria?
- (3) Any rationale for the ceramide effect being due to a single chain length (C16)?
- (4) In Fig 5 B why was complex I not included in the DKO MEFs?
- (5) A crucial piece of the argument in this manuscript is that the effects of PS inhibition on mitochondrial function is a result of failure to assemble the supercomplexes, resulting in the oxygen consumption defect. In Fig 6 B and C (BI and Myr) it looks to me that there is a general increase in the individual respiratory chain complexes that is driving what is apparently more supercomplex formation. Can this be ruled out? Maybe it would be useful to look at the total amount of each of the complexes on a DDM gel. If there are simply more complexes, this could in itself explain the rescue of the oxygen consumption defect.

Response to reviewers' comments

We would like to thank the reviewers for their comments and suggestions, most of which have helped to make the manuscript stronger. All substantive changes have been highlighted in yellow.

Comments of Reviewer #1

Major points:

1. Lipid analyses: the way in which lipid amounts are reported - as "molar mass over total moles of lipids analyzed" (mol %; e.g. in Fig 4) -is somewhat problematic. It is not at all clear what these numbers represent. A much more definitive number would be to give the nmoles of each lipid/mg cell protein so that the reader can directly assess whether the amount of that lipid is increased or not under a specific condition. It is also very unclear what is meant by "total moles of lipids analyzed" (mol %). Importantly it is not stated which lipids were included in this total value? Please either quote the data as nmole lipid/mg protein or give the 100% value of total nmoles of lipids analyzed.

We apologize for the lack of clarity. Mol% is a term regularly used by lipidomics facilities. In this context, mol% represent nmol/ μ l/ μ g of protein normalized by the total amount of lipids extracted and analyzed. We believe that representing lipid concentrations normalized only by protein content can sometimes be misleading, as this measurement does not account for changes in total amount of lipid per protein or technical variations during lipid extractions. We chose to use mol% rather than specific lipid concentrations in order to help us deduce the relative changes in lipid composition between control and mutant over the total lipid mass and protein content, but in truth, arguments can be made in support of both ways of representing the data.

We have therefore changed the graphs in the relevant figures to specify that our results represent nmol of lipid/ μ g of protein (or nmol lipid/ μ l) normalized by the total amount of lipids measured. In addition, we have included new supplementary figures showing the total nmols of lipids analyzed, normalized only by the protein concentration, in nmol/ μ g of protein.

2. Sphingolipid metabolism: it is obviously complicated to analyze sphingolipid synthesis and degradation in this situation. For example in Fig 4D, the incorporation of [3H]serine into ceramide and sphingomyelin is higher in the DKO than in controls. However, this result does not necessarily mean that ceramide synthesis or SM synthesis is increased in the DKO. First, it is not possible from this experiment to determine if synthesis is increased or if degradation is decreased in the DKO. Second, SM is a precursor of ceramide (SM degradation), and ceramide is a precursor of SM (SM synthesis). If radiolabel in SM were derived from radiolabeled ceramide, and if the radiolabel (and therefore specific radioactivity) in ceramide were higher in DKO than in control, the labeling of SM would automatically be higher in the DKO without an increase in SM synthesis. This is not the only complicating scenario. Thus, the wording about increased

sphingolipid synthesis needs to be carefully modified: there is in fact no evidence that the synthesis of ceramide or SM is increased in the DKO (see text page 8 etc). Nevertheless, the data on the SMase assays do indicate that SMase activity is higher in the DKO, and that the increase in ceramide is probably due to the increase in SMase activity rather than de novo ceramide synthesis.

We agree with the reviewer. In Figures 4D and 4E we show that DKO cells have increased *de novo* synthesis of sphingolipids and SMase activity. However, as Figures 5C and S4 show, our data suggest that the main source of the elevated concentration of ceramide is indeed upregulated SMase activity. We proposed that the increase in the *de novo* synthesis of sphingolipids is just the consequence of the need to replace the loss of SM due to its elevated hydrolysis by SMases. We have included and clarified this point in the text.

3. Myriocin expts Fig 6A: as a positive control for these expts, it would be very appropriate to confirm that amounts of sphingolipids (e.g. ceramide, SM as nmol/mg protein) in MAM and mito are indeed reduced by myriocin under the conditions used in these experiments.

Please see new supplementary Figures S6A and S6B.

Minor Comments

1. Why not include Fig S2C as a main Fig rather than a Supp Fig?

We thank the reviewer for this suggestion. We have moved the figure into the main text as Figure 2B.

2. Page 13, para 2. Further to the discussion on the role of phosphatidylserine the authors should consider adding some discussion of the role of the anionic lipid, PS, in mediating contacts between the ER and mitochondria [see Wu and Voelker (2004) JBC 279:6635 and a very recent paper from Prinz lab in J Lipid Res (2017)].

We have included these papers in the discussion.

Comments of Reviewer #2

Major points:

1. The different MEF cell lines (WT, PS1 KO, PS2 KO, DKO) are not of the same origin and may show protein expression level changes that go well beyond the genetic differences (PS KO). For this reason, it is state of the art to repeat at least the essential experiments with DKO cells re-transfected/reconstituted with either PS1 or PS2 or both to ensure the same background of the cells. This needs to be included.

We thank the reviewer for this suggestion. As shown in the new figures S4D, S4E, S4F and S4K, we transfected PS-DKO cells with plasmids expressing WT and mutant PS1 (A246E mutation) and found that WT, but not mutant PS1, was indeed capable of

partially rescuing the sphingolipid alterations and the upregulation of sphingomyelinase activity.

2. BACE inhibitors are prone to off-target effects, e.g. on cathepsin D, which is a main reason for several failed clinical trials with BACE inhibitors. Thus, key experiments in the study need to be repeated with a knock-down or knock-out/CRISPR of BACE1 or at least a BACE inhibitor with a different chemical structure.

Please see new Figures S3B (using a different BACE1 inhibitor on PS-DKO cells) and S3D (using BACE1-KO cells).

3. Please show lipid and mitochondrial changes in BACE1-deficient mice and in APP-deficient mice or in APP-transgenic mice. They do all have altered C99 levels compared to WT mice and thus should show changes similar to MEF cells, if the proposed function and mechanism are true. Importantly, show WT mice as a control. They are currently missing in figure S7.

Regarding the lipid changes, we have run lipidomics analysis of total homogenates and subcellular fractions from APP-KO mice and APP/APLP2-DKO MEFs. We do not see any significant differences in the ceramide or sphingomyelin levels in total homogenates or MAM membranes, and a decrease in phosphatidylserine levels in MAM membranes isolated from APP/APLP2-DKO cells, which lack C99 (please see the attached figures below). We believe that these data support our hypothesis, as these cells do not show sphingomyelinase activation when C99 is absent. On the other hand, many other lipid alterations are seen in the absence of APP and/or APLP2, but analysis of this phenomenon is outside the scope of the paper.

Lipid analysis of brain tissue from APP-KO mice

Lipid analysis of APP/APLP2-DKO MEFs

Regarding analyses in mitochondria, we have now also analyzed the lipid composition of mitochondria isolated from APP-KO mice and APP/APLP2-DKO cells, both of which lack C99. As in the case of total homogenates and MAM membranes, we believe that the absence of C99 in these samples impinges on sphingomyelinase activity. This is consistent with the finding of no significant differences in ceramide and sphingomyelin content in these cells vs controls (please see figure below).

Regarding Figure S7, control levels in the WT mice are shown by the dotted lines in panels A and B.

4. Many experiments are done in PS DKO cells, which (artificially) increase C99 levels. PS has around 100 different substrates and has additional functions in calcium signaling. Thus, I am not yet convinced that the artificial situation of PS-DKO is relevant to the situation in vivo.

We understand that many of our conclusions are based on assays done in PS-DKO cells, however, we have replicated all our main data in chemically treated cells, in fibroblasts from human patients, and in cells and tissues from animal models. We note that many of the assays are dynamic studies that require fresh tissue. Thus, the use of

human samples has been technically challenging, for obvious reasons. Nevertheless, the consistency of our results in the various patient and model systems that we have employed support our contention that the phenotypes that we have observed are real.

Finally, while it would be possible to perform a lipidomic analysis in brains from human AD patients and controls, we believe that this would be redundant, as the many studies already published in this area (some of which are cited in the text) describe the same lipid alterations that we have found in our samples, namely, increases in ceramide and subsequent decreases in sphingomyelin.

Lastly, in addition to replicating these changes, our goal in this manuscript has been to provide a potential mechanistic explanation for the lipid alterations seen not only *in vitro* but also *in vivo*, and the impact of these alterations on mitochondrial biology.

5. In mice, so far only one PS mutant is tested. This represents one form of familial AD, which makes up about 1% of all AD cases. 99% of AD are sporadic and do not involve PS abnormalities and probably also not C99 abnormalities. Yet, the title of the manuscript is about mitochondrial dysfunction in general in AD. Thus, you either need to test another AD model as well or change the title to better reflect the content of the manuscript.

We are willing to change the title to reflect this point, although we note that elevated C99 levels in sporadic AD have been reported numerous times in the literature. Those observations, together with our finding that the MAM deficits found in FAD are also present in SAD (see Area-Gomez et al, 2012, in which we analyzed 8 different FAD patients [with 6 different mutations in PS1, PS2, and APP], and 9 different SAD patients), reinforces our belief that, at bottom, FAD and SAD are fundamentally the same disorder from a pathogenetic point of view, and was the justification for the wording in the title.

Minor points:

6. In the introduction, the authors claim that increased C99 contributes to AD. This is an overselling of two previous publications and does not reflect the general state of the field.

We respectfully disagree with the reviewer. Although we cite 2 papers due to space limitations, numerous publications have shown the contribution of C99 to the disease (e.g. Forman et al., 1997; Kosik et al., 1999; Busciglio et al., 2002; Holsinger et al., 2002; Evin et al., 2003; Yang et al., 2003; Li et al., 2004; and Kim et al., 2016).

7. Fig. 1C: show C99 and C83 levels. It is known that under conditions of gamma-secretase inhibition there is more conversion of C99 to C83.

Please see previous Figure S2C, now Figure 2B.

8. Figure 2A: include an APP KO to ensure that the APP CTFs are specific bands. Also include - ideally in all gels - molecular weight markers. APP KO material will also help to ensure the specificity of the apparent AICD band in figure S2I. AICD is typically very difficult to detect.

We have transfected APP/APLP2-DKO cells with two different C99 plasmids. As shown in the WBs below, C99 signal is absent in untransfected DKO cells.

However, after transfection of APP/APLP2-DKO with these C99 plasmids, a band corresponding to C83 appears on the gel (compare with CTL cells exposed to DAPT). We do not understand why this is the case, although we note that we are not the first ones to see this C99 cleavage product.

We agree with the reviewer that AICD is quite difficult to detect, and in fact, we were not able to detect it in the western blot shown above. Nevertheless, the goal behind this figure is to show the increased C99 localization in MAM fractions isolated from brain tissues from PS1-KI mice compared to controls. That γ -secretase cleavage occurs in ER-MAM domains was already published by our group in 2009 (Area-Gomez, de Groof et al., 2009), and replicated by others (Schreiner, Hedskog et al., 2015)

9. Figure 2C: better separate the C99 and C83. Currently, it looks like one band instead of two. Indicate for the C99 gels which antibody was used. This is not clear from the methods section (N- or C-terminal antibody to C99).

We apologize for the lack of clarity. The antibody used is a C-terminal antibody to APP (Sigma # A8717).

Regarding Figure 2C (now Figure 2D), the higher concentration of C83 versus C99 makes the resolution of both bands in the upper part of the gradient quite challenging. Nevertheless, the goal of the gradient is to show how C99 can be found co-migrating with MAM markers. We tried to separate as much as possible both CTFs on the sucrose gradient and we believe that both bands are apparent in the inset shown below. The signal co-migrating with MAM markers is a single band corresponding only to C99, not C83.

Comments of Reviewer #3

1. It might be useful to show the Seahorse traces from Fig. 1 in the Supplemental data. Was the defect in CR compensated by an increase in ECAR? What about maximum uncoupled rate?

For Seahorse traces, please see new Figures S1P-T. Maximal uncoupled rate was reduced in PS mutant cells, as well as in cells treated with DAPT, indicating a lower spare respiratory capacity. We did not see any significant increases in ECAR during Seahorse analysis (data not shown).

2. In Figure 2A why is Lamp 1 completely localized to the mito fraction and in 2B?

Standard subcellular fractionations (as in Figure 2A) cannot successfully separate lysosomes, endosomes and mitochondria. For that reason, we decided to further analyze our sample by sucrose density gradients, as shown in Figure 2C. This more fine-grained approach allowed us to discriminate between endosomal, lysosomal, and mitochondrial and MAM markers to accurately localize C99.

why is VDAC in every fraction of the gradient? Is that true for other mitochondrial markers?

VDAC 1 is a mitochondrial marker also known to be enriched in areas of the mitochondria in contact with the ER (e.g., Prasad et al., 2015). We do not know the

reason why this marker is dispersed widely on the density gradient, although it is possible that this "dual" localization of VDAC in both the mitochondrial outer membrane and areas of the mitochondria in touch with the ER could affect its migration in this density gradient.

While the goal of this specific experiment was not to determine the localization of other mitochondrial markers, previous results in our lab have shown that mitochondrial markers with more "conspicuous" localizations, such as complex I subunits (inner membrane) or Tom20 (outer membrane), behave very differently in gradients compared to VDAC, and migrate to higher density areas of the gradient.

In 2D what is the explanation for the fact that the C99 positive foci are in many cases much larger than mitochondria?

We do not believe that this is the case. While the reason for the large C99 foci is puzzling, one possible reason is that the limited resolution of the confocal microscope is not sufficient to show individual C99 foci. For this reason, the accumulation of C99 at ER-mitochondria connections would appear as large foci instead of individual dots. The EM pictures showing "clusters" of presumably individual C99's (Figs. 4E and 4F) would support this view.

(3) Any rationale for the ceramide effect being due to a single chain length (C16)?

This is a very interesting point. We agree that C16 is the ceramide species where the changes seem more abrupt, although significant changes are also detected in C22, C24, and C24:1. We do not understand the reason behind these differences in ceramide species, although it is possible that C16 increases are not only the result of the upregulation of sphingomyelinase activity, but also the consequence of increases in the *de novo* synthesis by ceramide synthases 5 and/or 6.

(4) In Fig 5 B why was complex I not included in the DKO MEFs?

We apologize for this. During the development of this western blot we ran out of complex I antibody and used Tom20 instead.

(5) A crucial piece of the argument in this manuscript is that the effect of PS inhibition on mitochondrial function is a result of failure to assemble the supercomplexes, resulting in the oxygen consumption defect. In Fig 6 B and C (Bl and Myr) it looks to me that there is a general increase in the individual respiratory chain complexes that is driving what is apparently more supercomplex formation. Can this be ruled out? Maybe it would be useful to look at the total amount of each of the complexes on a DDM gel. If there are simply more complexes, this could in itself explain the rescue of the oxygen consumption defect.

We thank the reviewer for this nice suggestion. Please see new supplementary figure S6E.

Thank you for submitting your manuscript for consideration by the EMBO Journal. It has now been seen by the three original referees again, whose comments are enclosed.

As you will see, referee #1 and #3 now support publication, while referee #2 thinks that the *in vivo* support for your findings is not sufficiently compelling and does not answer the initial criticisms raised by this referee. I consulted further with referee #3 who thinks that embarking into additional mouse work would take up a considerable amount of time and who endorses publication without further *in vivo* data. Given this input, I would like to ask you to address the remaining concerns in a point-by-point response and by clearly outlining in your manuscript text what kind of additional *in vivo* support would be needed to better support the physiological relevance of your findings for AD.

I am therefore formally returning the manuscript to you for a final round of minor revision. Once we should have received the revised version, we should then be able to swiftly proceed with formal acceptance and production of the manuscript!

REFEREE REPORTS

Referee #1:

Summary

The authors have followed up on previous work (2009, 2012) in which they showed that gamma secretase activity resides in MAM, and that in models of AD alteration in activity of this protein modulates MAM function and the apposition between the ER and mitochondria. They now report that the 99 aa C-terminal fragment (C99) of APP is present in MAM. The authors also demonstrate that in models of AD the concentration of C99 in MAM is increased, resulting in increased degradation of sphingomyelin, consistent with the mitochondrial defects observed in AD.

Major and minor concerns

None. The authors have very carefully addressed all of my previous concerns. No additional suggestions for improving the manuscript.

Referee #2:

The authors have adequately addressed several of my previous points, although some of the experimental choices are surprising, such as the use of Gleevec, that is clearly not a BACE1 inhibitor, even though it affects (indirectly) BACE1 cleavage of APP. However, my major concern is still the lack of evidence that C99 contributes to the lipid and mitochondrial alterations under *in vivo* conditions and not only in cells. The (new) data provided correlate changes in C99 with alterations in lipid and mitochondrial metabolism, but they do not prove the causal link to C99 beyond cultured cell lines. For example, in the rebuttal letter the authors now report the absence of lipid changes in APP KO mice. This either speaks against a role of C99 in controlling lipid metabolism or could be interpreted in such a way that normal C99 levels need to be increased (for example with a PS mutation) in order to see the lipid changes. If the latter is true then APP tg mice (having WT and not mutated PS) should show similar lipid and mito changes as PS mutant mice. An alternative would be to cross BACE1-deficient mice with PS mutant mice. In this case, no C99 is anymore formed and the observed changes should be abolished. However, such an analysis is still lacking from the manuscript. This needs to be added to the manuscript or otherwise the data appear better suited for a more specialized journal.

Referee #3:

I think that the authors have now satisfactorily addressed the comments/questions of the reviewers and I have no further comments

2nd Revision - authors' response

18 August 2017

Response to Reviewer #2

The authors have adequately addressed several of my previous points, although some of the experimental choices are surprising, such as the use of Gleevec, that is clearly not a BACE1 inhibitor, even though it affects (indirectly) BACE1 cleavage of APP.

The use of Gleevac as BACE1 inhibitor tried to answer the previous suggestion from the reviewer: “BACE inhibitors are prone to off-target effects, e.g. on cathepsin D, which is a main reason for several failed clinical trials with BACE inhibitors. Thus, key experiments in the study need to be repeated with a knock-down or knock-out/CRISPR of BACE1 or at least a BACE inhibitor with a different chemical structure”

We could not find any commercially available BACE1 inhibitors that did not have off target effects. Therefore, instead of trying another chemical inhibitor, we decided to use Gleevec, based on data published by the groups of Victor Bustos and Paul Greengard, that demonstrates that Gleevec shifts APP cleavage towards the non-amyloidogenic pathway (Netzer et al, PNAS, 2017). Whether directly or indirectly, the goal of our experiments is reduction in C99, not inhibition of BACE1 (which is merely a means to that end).

We also added data from BACE1 knock-out cells. Some of the previously suggested CRSPR experiments were extremely interesting, but the production, characterization and authentication of these cells would take longer than the allowed resubmission time.

However, my major concern is still the lack of evidence that C99 contributes to the lipid and mitochondrial alterations under in vivo conditions and not only in cells. The (new) data provided correlate changes in C99 with alterations in lipid and mitochondrial metabolism, but they do not prove the causal link to C99 beyond cultured cell lines.

While we believe that our data in AD patients and in animal models support the role on C99 in inducing these lipid changes, our results mainly show a correlation between MAM-localized C99 and lipid and mitochondrial alterations. However, this correlation does not negate that the fact that C99 is playing a key role in the induction of these phenotypes, as abrogation or reduction in C99 production reverses those phenotypes. Whether this effect is mediated directly by C99 at the MAM or indirectly via a yet-unidentified MAM-localized C99-binding partner is still under investigation.

For example, in the rebuttal letter the authors now report the absence of lipid changes in APP KO mice. This either speaks against a role of C99 in controlling lipid metabolism or could be interpreted in such a way that normal C99 levels

need to be increased (for example with a PS mutation) in order to see the lipid changes.

We apologize for not being clearer. The APP-KO cells and mice show many lipid changes, but we only included those related to the phenotype under study (i.e., sphingolipid metabolism). As mentioned by the reviewer, we indeed suggest that MAM-localized C99 needs to be increased (e.g. via a PS mutation) to trigger sphingomyelin hydrolysis and the subsequent elevation in ceramide. This does not occur in APP-KO where C99 is, of course, absent. Nevertheless, these animal models display many other lipid changes, but we feel that reporting/analyzing those changes is outside of the scope of this paper.

Of note, our data showing that elimination of C99 (by BACE1 inhibitors) rescues lipid disturbances in AD cells, suggest that C99 has a causative role in the induction of these lipid alterations. However, we used this approach (i.e. BACE1 inhibition) as a proof of principle, and we do not believe, nor do we suggest, that total inhibition of C99 production is innocuous.

If the latter is true then APP tg mice (having WT and not mutated PS) should show similar lipid and mito changes as PS mutant mice.

We agree with the reviewer. Our data show that C99 needs to be elevated in order to cause these lipid alterations. These elevations in C99 are detected in familial cases of AD (due to either mutations in PS's or APP), Down syndrome cases due to APP triplications, and SAD due to currently unknown reasons.

An alternative would be to cross BACE1-deficient mice with PS mutant mice. In this case, no C99 is anymore formed and the observed changes should be abolished. However, such an analysis is still lacking from the manuscript. This needs to be added to the manuscript or otherwise the data appear better suited for a more specialized journal.

While we will be happy to do these experiments, we would not be able to provide these data in the time frame of the resubmission process.

Thanks for submitting your revision to the EMBO Journal and for sorting out the last few details. I am very pleased to accept the manuscript for publication here.

Corresponding Author Name: Estela Area-Gomez

Journal Submitted to: EMBO journal

Manuscript Number: EMBOJ-2017-96797R